# How much time to figure out how to get where? Route planning and subjective stress under time pressure

Paul E. Plonski[1¤]*, Prsni Patel[1], Kathryn L. Ossenfort[1], Holly A. Taylor[1,2], Tad T. Brunyé[2,3], Heather L. Urry[1,2]*

1 Department of Psychology, Tufts University, Medford, Massachusetts, United States of America, 2 Center for Applied Brain and Cognitive Sciences, Tufts University, Medford, Massachusetts, United States of America, 3 U.S. Army DEVCOM Soldier Center, Natick, Massachusetts, United States of America

¤ Current address: Department of Psychology, Swarthmore College, Swarthmore, Pennsylvania, United States of America

* paul.plonski@swarthmore.edu (PEP); heather.urry@tufts.edu (HLU)

**Data Availability Statement:** The pre-registration can be found at https://osf.io/vc53a. Materials, de-

## Abstract

From a daily commute to military operations in hostile territory and natural disaster responses, people frequently move from place to place. Cognition (e.g., wayfinding) occurs in conjunction with behavior (e.g., locomotion) to facilitate spatial navigation–intentional movement through space. People often use maps to plan routes, which is part of wayfinding. Time pressure is common during navigation, even during route planning, for example from time constraints (e.g., a deadline), waiting periods (e.g., technological problems), or imposed urgency (e.g., someone tells you to hurry up). Route planning requires knowing where to go, determining how to get there, and managing transient stressors that can influence performance. Across cognitive and behavioral domains, time pressure is often conceptualized as a stressor and examined with a single operationalization. As a result, we do not know to what extent time constraints, waiting periods, and imposed urgency independently or interactively a) contribute to the sense of subjective stress, and b) impact spatial performance. Our work addressed these knowledge gaps using a computerized spatial task that centrally involved planning and tracing routes on maps. We describe this new methodology for studying route planning and demonstrate experimental effects of urgency messaging on increased subjective stress and decreased time between map presentation and first click (planning time). When participants took longer to plan or drew longer routes, they reported greater subjective stress. Results carry implications for the design and implementation of time pressure manipulations, route planning in stressful conditions, and mitigating or optimizing stress effects on performance.

## Introduction

In daily life and extreme situations, the ability to successfully navigate one's environment is essential. Spatial navigation requires both wayfinding (planning and decision making) and

identified data, and analysis scripts can be found at
https://osf.io/mcbgn/.

**Funding:** This research was sponsored by the U.S.
Army DEVCOM Soldier Center and was
accomplished under Cooperative Agreement
Number W911QY-19-2-0003. The views and
conclusions contained in this document are those
of the authors and should not be interpreted as
representing the official policies, either expressed
or implied, of the U.S. Army DEVCOM Soldier
Center, or the U.S. Government. The U.S.
Government is authorized to reproduce and
distribute reprints for Government purposes
notwithstanding any copyright notation hereon.
The sponsors or funders did not play any role in
the study design, data collection and analysis,
decision to publish, or preparation of the
manuscript.

**Competing interests:** The authors have declared
that no competing interests exist.

locomotion [movement of a body, 1]. Under the umbrella of navigation, the term *wayfinding*
was coined at least as early as 1960 by Kevin Lynch and more recently defined as "the process
of determining and following a *path* or *route* between an origin and a destination" [2], p. 6).
Although the definition of wayfinding can encompass movement through space [3], wayfind-
ing is more often taxonomically separated from locomotion [4, 5]. One aspect of wayfinding is
route planning, an activity often undertaken with a map that provides access to maximal spa-
tial information. There are a myriad of stressors which could affect the process of spatial navi-
gation, for any person, from the early planning stages to the locomotion from place to place.

Although physical capacities, spatial-cognitive abilities, and available information are essen-
tial to navigation, whether the context is stressful also matters [6, 7]. For example, a daily com-
mute may be an incredibly complex cognitive process, but easily accomplished by most
individuals. If, however, an accident causes a traffic jam or road work requires modification to
a planned route, the trip may become difficult and provoke psychological stress. Psychological
stress can be defined as the state elicited following appraisals by an individual of their potential
inability to cope with the demands of a situation. Psychological stress can be framed as emo-
tionally positive (e.g., challenge or engagement), but can also have negative emotional valence
[e.g., harm or threat appraisals, 8]. Stressful contexts may lead to stress responses during
wayfinding.

Sustained psychological stress over extended periods of time (e.g., chronic stress) has long
been theorized to have negative consequences [9], although short-term stress responses (e.g.,
acute stress) can be adaptive [10]. Since wayfinding tends to be time-limited, stressors related
to wayfinding are most often acute, such as an escape from a fire emergency [11]. One adaptive
response of acute psychological stress is enhanced allocentric spatial processing–thinking spa-
tially from a perspective outside one's own body [12, 13]. Acute affective responses may be par-
ticularly beneficial at low levels of arousal and in novel environments [14]. While adaptive,
stress can also be costly; for example, threat of electric shock has been shown to impair allo-
centric spatial memory [15], affect neural underpinnings of spatial representations, and could
encourage taking familiar routes over possible shortcuts [16].

In laboratory settings, researchers often induce acute stress with time pressure by limiting
the actual (i.e., objective) or perceived (i.e., subjective) amount of time available to do some-
thing [17]. Using time constraints to create time pressure has been referred to as temporal
stress [18]. To generate time pressure, researchers often estimate the average time a task will
take (for a person or a group) and then allot less time than typically required [e.g., 19, 20]. Like
stress from electric shocks [16], under time constraints, participants favored familiar paths
over shortcuts [19]. In general, time constraints can speed up performance [21], speed up
information processing, and alter specific cognitive strategies [22].

Time constraints might be considered the prototypical type of time pressure, but acute
stress can also be induced by making people wait, although this operationalization is uncom-
mon. In a study of forced waiting periods on performance speed, waiting sped up performance
in an industrial-organizational context [23]. In the car, waiting periods such as long red lights
have been incorporated in studies of emotion [24]. In a study of emotion during wayfinding,
when participants were told that they were driving and running late, they had more intense
feelings of impatience during a stop [25]. Therefore, being forced to wait might be considered
a type of time pressure, especially when there is an overall time constraint.

Both time constraints and waiting periods induce time pressure by decreasing the amount
of time available to complete a task. However, limiting the amount of time available is not uni-
versally associated with negative emotions and stress [26]. For a context to inculcate stress,
there may need to be a subjective sense of urgency, such as when a passenger tells a driver to
"Hurry up!" [27]. Meta-analyses suggest that time pressure induced by a sense of urgency (e.g.,

instructions to perform as quickly as possible) or temporal constraints (e.g., deadlines) can increase speed and decrease accuracy in cognitive and perceptual tasks [28]. Urgency messaging is different from time constraints and waiting periods because it does not require altering the measured amount of time available to complete a task; its effects are subjective.

In sum, the literature reviewed above suggests that acute stress can enhance allocentric spatial processing [11–13], and that time pressure [19] and stress associated with electric shocks [16] can predispose taking familiar paths over shortcuts in previously learned virtual environments. However, it is unclear to what extent psychological stress can be provoked by temporal constraints, waiting periods, or an externally imposed sense of urgency. Furthermore, the effect of time pressure on planning a route in a new environment is essentially unknown. Although navigating a novel environment might occur less frequently than, say, a daily commute or running errands, there are numerous situations in which navigating a new environment is important. For example, people often plan many routes while visiting a new city. Routes are also planned for military or humanitarian missions into new environments. When the environment is new, certain factors that are relevant to navigation, such as spatial memory, may be less important than when an environment has been learned. Moreover, new environments can induce anxiety [29]. Because the psychological processes are different in new versus familiar environments, we might also expect different effects of time pressure. Recent studies of stress and time pressure during wayfinding focused on learned environments [16, 19], rather than new environments. Overall, there is a dearth of knowledge on time pressure or stress and route planning, despite substantial research on time pressure and stress on performance, and even performance in spatial domains.

The primary purpose of the present study was to determine to what extent temporal constraints, waiting periods, and a sense of urgency have independent and interactive effects on subjective stress and route planning in new environments. We designed an online task to assess route planning performance using static, full-view, 2-D maps. As part of the task, we imposed global time constraints (35 vs. 25 minutes) and varied the average amount of time lost during intermittent waiting periods (10 vs. 50 seconds) between subjects. We also randomly imposed urgency at the trial-level within subjects. The task allowed us to measure psychological stress and route planning in a manner that required map reading, decision making, and precision motor performance. Our procedures are reminiscent of a navigational situation like the travelling salesperson problem [e.g., 30, 31], but with only one or two target locations. We obtained three primary metrics from the task: 1) the number of trials completed, 2) the time between stimulus presentation and the first click, which we refer to as planning time, and 3) the length of each route. We also measured subjective stress at the end of each trial.

We had several hypotheses. First, we expected that time constraints, waiting periods, and urgency messaging would affect subjective stress and route planning performance. Specifically, we hypothesized that larger time constraints (i.e., having only 25 minutes) and longer waiting periods (i.e., having to wait 50 seconds on average), both of which decreased the effective amount of time on task, would decrease the "amount" of performance, resulting in fewer routes planned compared to smaller time constraints (35 minutes) and shorter waiting periods (10 seconds on average) (H0, labeled H6 in preregistration). In addition, we hypothesized that the three higher-time pressure conditions (having less total time, longer average waiting time, and exposure to urgency messaging) would be associated with greater subjective stress (H1) and shorter planning time (H2) compared to the lower-time pressure conditions. Finally, because time pressure generally increases speed and decreases accuracy, we expected that the higher-time pressure conditions would result in longer routes than the lower-time pressure conditions (H3). However, because the effects of acute time pressure in new map

environments have not been studied extensively, we intentionally remained open to opposite or null effects of all three experimental manipulations.

Second, we expected that trial-level subjective stress–that is, stress reported after tracing each route–would be associated with performance metrics. Specifically, we hypothesized that trial-level subjective stress would be associated with decreased planning time (H4) and increased route length (H5).

Finally, outside of wayfinding contexts, some researchers have emphasized individual differences in time urgency when considering time pressure and stress responses [32]. Thus, we were interested in determining to what extent we would observe individual differences in trial-level outcomes. We hypothesized there would be substantial individual differences in subjective stress and performance outcomes (H6, labeled H7 in our preregistration). Discovery of individual differences would open the door to future investigations of between-subjects variables that contribute to time pressure effects on these outcomes.

## Method

### Statement of transparency

The research questions, method, and planned analyses (including exclusion criteria) for this study were registered prior to data collection. Participants were recruited and completed the study procedures between August 23, 2021 and September 8, 2021. Written consent was provided by responding to the consent form in an online survey before taking part in any research procedures. Data, materials, and analysis scripts are available in our Open Science Framework project. Deviations from the preregistered plan are noted. The preregistration did not specify how we would mean-center subjective stress. In the reported results, we used two variables for subjective stress to separately assess within-person and between-person variability. The research protocol number STUDY00001374 was approved by the Tufts University Social, Behavioral and Educational Research Institutional Review Board. The protocol was also approved by the U.S. Army DEVCOM Soldier Center Human Research Protections Office.

### Design

We used a 2 x 2 x 2 mixed factorial design. There were two between-subjects factors, time constraints and waiting times, and one within-subjects factor, urgency messaging. Time constraints had two levels; participants had 35 (more time) or 25 (less time) minutes to complete the route planning task. Waiting times also had two levels; participants had waiting periods averaging 10 (short waits) or 50 (long waits) seconds during the task. Lastly, urgency messaging had two levels; each trial randomly presented "not urgent" (not urgent) or "urgent, hurry up!!!" (urgent) messages. There were three trial-level outcomes, subjective stress, planning time, and route length, and one participant-level outcome, the number of routes planned (i.e., the number of trials completed). We motivated participants to complete as many trials as possible without tracing errors by providing a small bonus, $0.05, per trial completed.

### Participants

We employed Prolific (Prolific, Oxford, UK) services to recruit 300 English-speaking adults in the United States (US) in 2021. The sample was representative of the United States by proportions of gender, age, and ethnicity (specifically, Asian, Black, Mixed, Other, and White) based on the 2010 US Census. The sampling method and sample size were determined primarily by the available funds, task duration, and the intention to collect a diverse sample from which we could generalize results. Sampling from the English-speaking United States population

reduced potential variability associated with language or culture between countries, which was helpful because the study was not large enough to examine cultural variability empirically.

Participants whose data were included ($N$ = 226; see Exclusions section below) reported female ($n$ = 125), male ($n$ = 100), or nonbinary ($n$ = 1) genders, and their median age was 40 years ($M$ = 42.35, $SD$ = 16.78, $min$ = 18, $max$ = 75). Almost all participants responded to the open-ended question "What word or phrase best describes your race and/or ethnicity?" with at least one racial or ethnic group (Asian and/or Asian American = 22, Black and/or Black/African American non-Hispanic = 7, Hawaiian = 1, Hispanic = 12, Human = 1, Mixed or Mixed Race or Multi-Ethnic = 5, Native American = 1, Other = 2, White and/or White/Caucasian American non-Hispanic = 151). The Asian and/or Asian American classification included East Asian = 1, Filipino American = 1, Indian = 1, Korean = 1, and South Asian = 1. The Hispanic classification included Latino = 1, Mexican = 1, White Latina = 1.

Participants were paid a base rate of USD$8.25 for participation in a 75-minute study. To encourage participants to complete as many trials as possible, they were informed that they would receive a performance bonus (USD$0.05) for each map completed successfully. They were informed of the bonus in the initial study advertisement and in the informed consent procedure. When data collection was completed, all participants who actively participated in the route planning task (i.e., they drew routes and appeared to follow some paths) were awarded a bonus to bring their total compensation to USD$12.00.

## Materials

**Map stimuli.** We created 120 map stimuli for this task by manually searching for suitable locations on Google Maps and screenshotting the area (https://osf.io/w36xe/). The latitude and longitude for each stimulus in decimal degrees are available in S1 Data. We intended to create a set of stimuli from which we could generalize results, which influenced the choices we made while designing the stimuli. We created multiple categories of map, based on the placement of the points, to be sure there was variation in the features of the routes that participants had to plan and trace. The location for each map stimulus was relatively random, but they were generally clustered in areas across the United States that were likely unfamiliar to most participants. For each cluster, we looked for a small town or urban or suburban area that we thought would be unfamiliar to most participants. We looked for levels of complexity in roads and landmarks when visually selecting locations. We ensured that there were multiple ways to get from place to place on every map.

Three categories of map stimuli were designed to improve the generalizability of the effects of time pressure across routes with one section and routes with two sections. The map stimuli were thereby categorized by the arrangement of points (starting point, ending point, and for some maps, a waypoint). One set of stimuli contained only a starting point and an ending point (type 1, $n$ = 41). The remaining stimuli contained a waypoint, in addition to a starting and ending point. Of those with a waypoint, one set of stimuli (type 2, $n$ = 44) could be considered to have two unique sections, one section from the start to the waypoint and the other from the waypoint to the end. The last set of stimuli (type 3, $n$ = 35) was marked with the same start and end location, so that any route needed to return to the starting location after visiting the waypoint.

When creating each stimulus, we first selected a starting point for the map, ensuring that there was variability in the cardinal direction of the start relative to the center across the stimuli. We selected locations for the other points to ensure variability in the cardinal direction of travel across stimuli. We created and documented at least two routes that were relatively similar in length for each stimulus. There were differences in the potential routes that could

involve decisions such as where to turn if multiple turns were possible, or whether to stay closer to the edges of the map. Stimuli (768 pixels x 768 pixels) were converted to arbitrary standardized units from -0.5 to 0.5, including a border around the map that was equal on the vertical and horizontal dimensions. Therefore, units remained the same regardless of the physical size or resolution settings of the monitors used by participants, or the scale of any specific stimulus.

**Route planning task.** We built a route planning task (available at https://gitlab.pavlovia. org/ebbl.tufts/route_planning_task_time_pressure) using the PsychoPy Builder interface (Peirce et al., 2019) and administered the task online as a PsychoJS experiment (Bridges et al., 2020). Participants planned routes and then used their mouse or trackpad to trace the routes on the digital map images described above. There were two primary performance outcomes conceptualized *a priori*, 1) the time between stimulus presentation and the beginning of the tracing phase, termed planning time, and 2) the two-dimensional distance of the route traced using the mouse or trackpad, termed route length.

We provided detailed instructions about how to complete the task, including our performance expectations. We used on-screen text instructions, as well as three practice trials. Participants were instructed to trace routes on the roads and only on the roads, by holding down a mouse or trackpad button while moving the cursor. Instructions emphasized tracing as many routes as possible without making tracing errors.

To begin tracing, after a stimulus was presented, participants started the tracing phase by pressing 's' on the keyboard. During the three practice trials, after the "s" was pressed, participants were asked whether the stimulus contained a waypoint. Veridical feedback was provided immediately upon response. This question ensured that participants were familiarized with all trial dynamics, including the identification of any waypoint. When participants had completed their route, they finished the trial by pressing 'f' on the keyboard.

To ensure that all participants knew the desired degree of precision motor performance in this tracing task, orange "bumper lines" depicted an appropriate amount of deviation from the marked roads in the second practice trial (see Fig 1). We reminded participants of our performance expectations on the third practice trial in the form of a single short section of similar bumper lines near the end of a route that had no alternative paths. The real trials showed no bumpers and participants could, in theory, trace a route anywhere on the maps.

Participants started each trial with a mouse click on the lower portion of a home screen with two circles, one on the left and one on the right. Instructions to click the circle on the right served as a trial-level attention check (any click near the bottom would begin the trial and any click on the right-hand side counted as successful). After beginning the trial, participants saw a 500ms message designating the trial "urgent" or "not urgent". Then, a map was presented with a starting and ending location marked; some maps also had a waypoint. Completion of a trial included two phases, as described above, a planning phase and a tracing phase. Participants were instructed to take as much time as they needed to study the map and mentally plan a route (planning phase) The planning phase included the time that it took the participant to identify the starting location, destination, and any potential waypoints. They were instructed to move their cursor to the starting location after planning and press 's' on the keyboard button to start the tracing phase. They traced the route by holding the left button on a mouse or trackpad and pressed 'f' upon arriving at the end point. They then rated their level of subjective stress.

**Subjective stress.** Participants rated their level of subjective stress ("How stressed were you during the last trial?") on a continuous visual analogue scale (VAS) after each trial. The VAS ranged from 0, *Not at all stressed*, to 100, *Extremely stressed*, which was rounded to the nearest integer for analysis on a 101-point scale. Participants also responded to a similarly

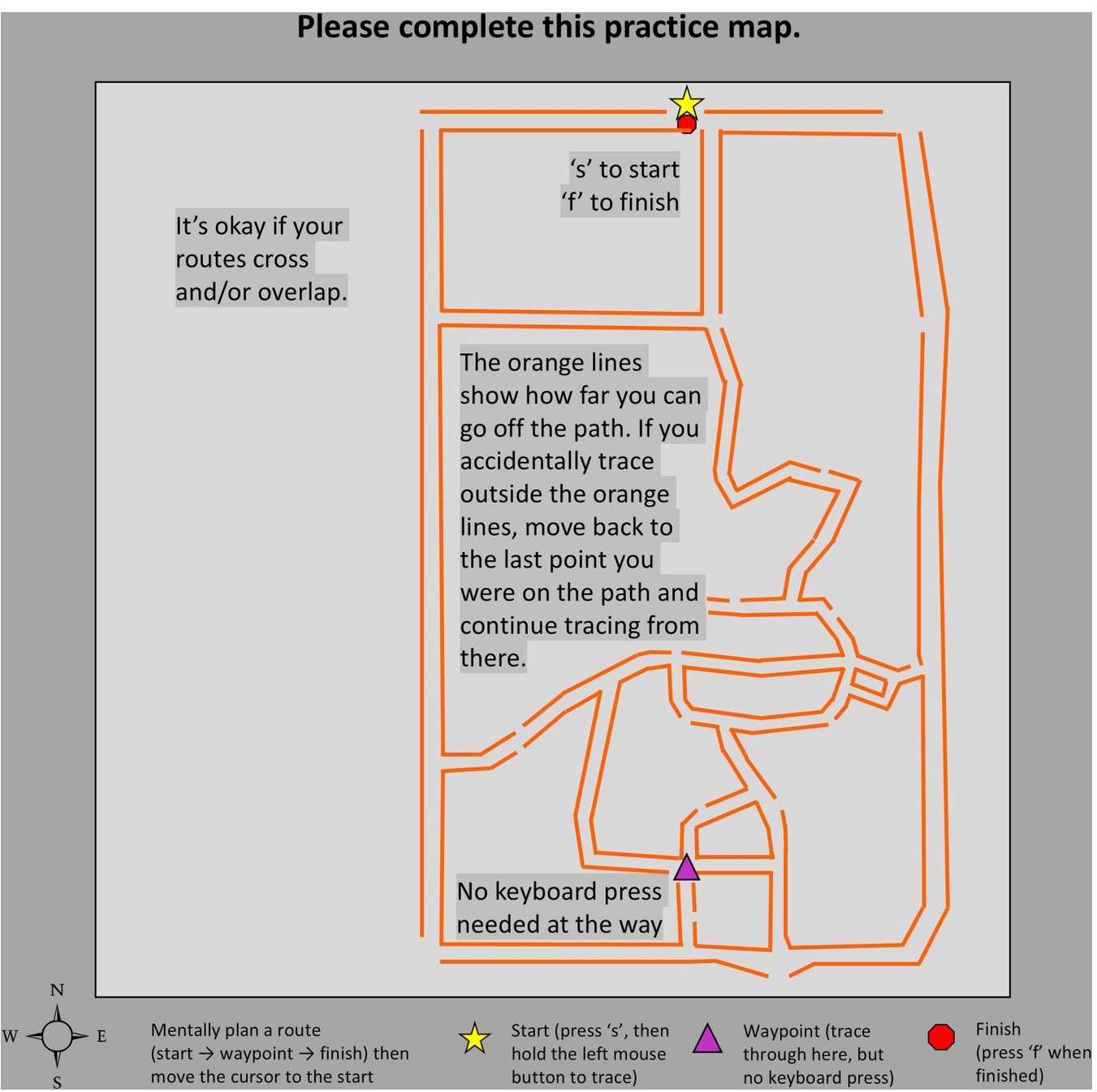

**Fig 1. Practice trial #2.** A map stimulus was presented at the start of each trial. The orange "bumper" lines represented the degree to which the traced route was "allowed" to deviate from the path before it was considered an error. Bumper lines were only included on the second practice trial (shown here) and a single section on the third (not shown). The route traced by the participants appeared to them as a dark blue line as they moved the cursor with either the mouse button or the trackpad button held down. Note that the map seen by participants (centered at latitude and longitude [39.2012283, -86.5338801] in decimal degrees), which was created using Google Maps, has been removed from this figure for copyright compliance.

scored VAS for subjective stress before beginning the practice trials ("How stressed were you so far?"), after each of the practice trials (i.e., "How stressed were you during the last practice trial"), and at the end of the task (i.e., "How stressed were you during the whole task").

**Subjective time pressure (Manipulation Check).** Post-task, participants reported subjective experience of time pressure on three continuous VASs from 0 (*No pressure*) to 100 (*Extreme pressure*) by responding to the questions: "How much time pressure did you experience during the whole task?", "How much time pressure did you experience on the "urgent" trials?", and "How much time pressure did you experience on the "not urgent" trials?". Each score was rounded to the nearest integer for analysis on a 101-point scale.

## Power analyses

Two sets of simulations were conducted, each with a different outcome variable. The first simulations used only the between-subjects manipulations and the second added the within-subjects manipulation. Previous effect sizes were not available for the experimental manipulations, so hypothetical effect sizes were used. The general conclusion was that we had adequate power to detect main effects of experimental manipulations under most simulated conditions but only adequate power to detect an interaction effect between the between-subjects manipulations under specific conditions. First, we simulated a Gaussian-distributed planning time outcome. Each between-subjects time pressure manipulation was set to have a medium main effect (Cohen's $d$ = 0.50). For the chosen sample size ($N$ = 300), power analysis by simulation ($n_{\text{simulations}}$ = 1000) suggested adequate power (86%) to detect a medium-sized main effect of a between-subjects manipulation. Power decreased for small or interaction effects, such as a small interaction ($d$ = 0.25) between the between-subjects effects (total time and wait time) that was underpowered at 17%. Power analyses were preregistered and updated after preregistration, but before data collection.

Then, simulations were conducted to examine power to detect medium effects ($d$s = 0.50) on a Gaussian-distributed route length outcome. The second power analysis included the within-subjects manipulation of urgency. All effects which included the within-subjects manipulation of urgency had high power (>99%), whereas the power to detect an interaction between the two between-subjects manipulations was only adequately high (>80%) when all main effects and interactions were included and of medium size in the simulation. In general, power to detect any interaction between total time and wait time was at or below 50%, which suggested caution interpreting that interaction. Further caution is suggested as the final sample size after excluding participants who did not follow instructions was smaller than the anticipated sample size.

## Procedure

Participation occurred online in a time and place of the participants' choosing. After signing up for a study titled "Wayfinding Under Time Pressure" on Prolific, participants were redirected to Qualtrics (Qualtrics, Provo, UT) where they were asked to provide informed consent. Notably, the consent form included information about the performance-based bonus payments and the statement "if you are having an exceptionally bad day, or if you are particularly busy right now, you may not wish to participate". The form also included information about the two total time conditions (35 or 25 minutes), but the other time pressure manipulations were not described at the time of consent for simplicity. Participants who consented were randomly assigned a condition for each of the two between-subjects factors and asked to respond to baseline self-report questionnaires (see S1 Text) before being redirected to Pavlovia (Open Science Tools Ltd, Nottingham, UK) to complete the route planning task.

Upon redirection to Pavlovia, participants received instructions and practiced the route planning task. After practice, they were informed of the waiting periods (but not the existence of different waiting period conditions) and that some trials would be designated urgent. Then,

**Table 1. Between-subjects time pressure conditions.**

|  | More Time/ Short Waits | More Time/ Long Waits | Less Time/ Short Waits | Less Time/ Long Waits |
|---|---|---|---|---|
| Total Allotted Time | 35 mins (2100s) | 35 mins (2100s) | 25 mins (1500s) | 25 mins (1500s) |
| Total Wait Time | 140s | 700s | 100s | 500s |
| Wait Times (seconds) | 2, 2, 6, 6, 8, 8, 10, 10, 12, 12, 14, 14, 18, 18 | 42, 42, 46, 46, 48, 48, 50, 50, 52, 52, 54, 54, 58, 58 | 2, 2, 6, 8, 10, 10, 12, 14, 18, 18 | 42, 42, 46, 48, 50, 50, 52, 54, 58, 58 |
| Wait Time (mean) | 10s | 50s | 10s | 50s |
| Effective Task Time | 32 mins + 40s (1960s) | 23 mins + 20s (1400s) | 23 mins + 20s (1400s) | 16 mins + 40s (1000s) |
| Sample Size Analyzed (*n*) | 57 | 57 | 54 | 58 |

Timing parameters and sample size for the between-subjects experimental conditions of the route planning task. Each column contains the specifications for one of the four cells of the between-subjects portion of our design. More Time/Short Waits = low time pressure for total time and wait time; More Time/Long Waits = low time pressure for total time and high time pressure for wait time; Less Time/Short Waits = high time pressure for total time and low time pressure for wait time; Less Time/ Long Waits = high time pressure for total time and wait time. Total time was either 35 or 25 minutes. The average wait time was either short or long, but the total number of seconds was variable based on the total time condition and the speed of performance. The number of seconds was randomly selected from a set of possible times, shown for each condition in row three. The effective task time is the total time minus the total waiting time, assuming that no single map took more than 150 seconds to complete. The sample size is the number of participants in each group, after all exclusions.

participants completed the route planning task by random assignment to their between-subjects conditions, following the instructions described in the *Route Planning Task* subsection above.

The first between-subjects manipulation, conveyed by a countdown timer, was the total amount of time allotted (35 vs. 25 minutes). The timer appeared at the top of the screen for all participants when the task began. The second between-subjects manipulation was the average length of the waiting periods (10 vs. 50 seconds). Each waiting period occurred after completion of the subjective stress rating (between trials), approximately once every 150 seconds. Participants allotted 35 minutes experienced 14 waiting periods and those allotted 25 minutes experienced 10 waiting periods (see Table 1 for details). Due to the intermittent spacing of the waiting periods, if a participant took longer than 150 seconds to complete a single trial, they experienced one less waiting period.

The within-subjects manipulation was whether each trial was designated "Not Urgent" or "Urgent" with 50% probability (see Fig 2). After the participant started the trial, the map stimulus was preceded by a 500ms cue designating the upcoming trial as not urgent or urgent. After the cue, one of 120 map stimuli was presented by random assignment without replacement for that participant.

After completing the route planning task, participants were redirected to Qualtrics to complete a series of self-report questionnaires including questions about financial motivation and a short stress state questionnaire. Then, participants were directed back to Pavlovia to complete a multi-source interference task and additional self-report items (see S1 Text). Following the interference task, participants were directed to Qualtrics to provide demographics and for debriefing and finally, to Prolific to confirm participation for payment.

## Exclusions

Using preregistered criteria, we excluded participants from all analyses if they reported being under age 18 years (*n* = 0), resided or were currently located outside of the United States (*n* = 0), or reported having participated previously (*n* = 4, 1%). We excluded six participants (2%) who did not consent to their data being included following debrief (the remaining data,

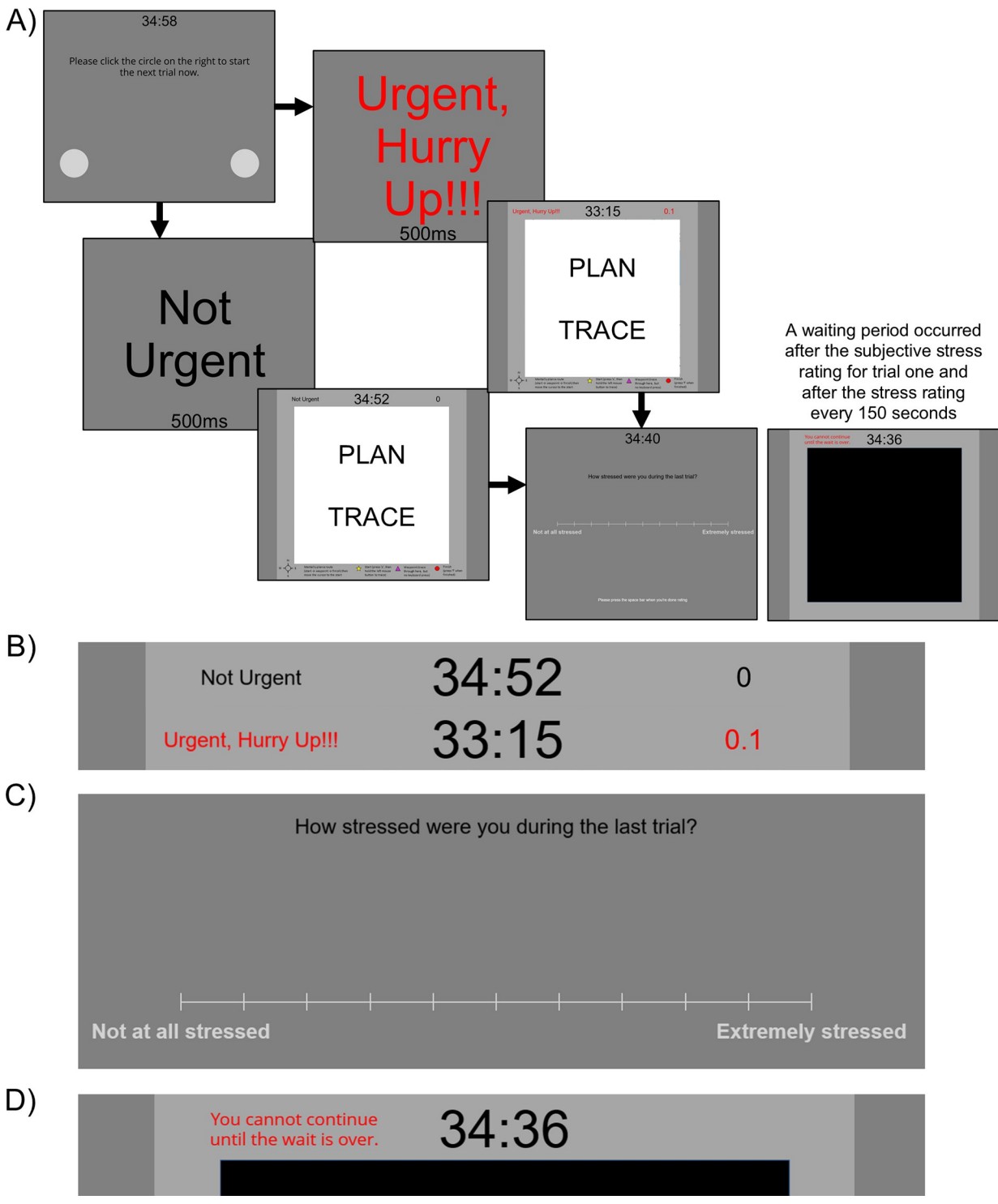

**Fig 2. Route planning task trial diagram.** Panel A) depicts a trial in the route planning task. Participants had 25 or 35 minutes to complete as many trials as possible without tracing errors. After each home screen, there was a 500ms urgency messaging condition cue before map stimulus presentation. The map remained on the screen throughout the planning and tracing phases. As depicted in Panel B), time pressure was conveyed throughout the trials and waiting periods. A black countdown timer was present after instructions and practice trials, for all participants throughout the entire task, in the top center of the screen. Urgency messages were presented in the upper left and for urgent trials, a red "count-up" timer in increments of 0.1s was active in the upper right. Panel C) depicts the subjective stress rating scale, which appeared after each map stimulus. Panel D) depicts the top portion of a waiting period screen. See Table 1 for details on waiting period timing. Note that maps seen by participants, which were created using Google Maps, have been removed from this figure for copyright compliance.

$N$ = 290, are publicly available). Prior to analyses, five participants (2%) were excluded who self-reported touchscreen use after instructions to use a mouse or trackpad, and one who used a touchscreen but did not self-report touchscreen use. The primary preregistered exclusion criteria was at the participant level. who did not follow instructions on at least five trials (i.e., plan time greater than 60s, did not click the correct button to start the trial, or did not attempt to follow the paths; $n$ = 57, 19%). See S1 Text for details on the performance review that we used to determine whether participants attempted to follow the paths.

Data from one participant (<1%) were excluded because they spent most of the task on one trial start screen (1361 out of 1400 seconds); this was not a preregistered exclusion criterion (this pattern suggested that the participant intentionally did not participate in the task and instead simply waited for the countdown timer to near the end; the longest time any participant spent on a trial start screen on an included trial was 254 seconds). After participant-level exclusions, date from $N$ = 226 participants were available for analysis.

Then, trials were excluded if the participant did not attempt to follow the path (18 trials, <1%) or planning time was greater than 60 seconds (127 trials, 0.02%), in accordance with the preregistered plan. Trials with long planning times were excluded because they indicated to us that there was a lack of active participation at that time. Although not a preregistered criterion, a single trial with a plan time rounded to two decimals equal to zero was also excluded. The trial with zero plan time was excluded because it defied assumptions of a response time measure in that the outcome must be greater than zero. Trials begun with an incorrect click were excluded (2 trials, <1%) because this suggested a substantive lack of attention to the instructions at that moment. Trials that were excluded for short plan time, long plan time, or incorrect starts were still included in the calculation of the number of trials completed (unlike trials in which participants appeared to violate the instructions). Even if there was a substantial distraction that invalidated a metric at a particular time during the trial, a participant could still have planned and traced a route in good faith.

## Analysis plan

Analyses were conducted with R 4.3.0 [33], visualizations with ggplot2 [34], and tables with sjPlot [35]. Confirmatory analyses used linear mixed-effects regression models with the lme4 package [36] and lme4 extension lmerTest [37]. We also used the performance [38] and insight [39] packages to examine the models. We modified the preregistered manipulation checks due to redundancy and conducted a single mixed ANOVA on the post-task subjective time pressure ratings with the ezANOVA package [40].

The models used to test experimental effects on trial-level outcome variables followed Eq (1); outcomes were modeled with three fixed-effect experimental conditions and their interactions, plus two random intercepts, participant ($\mu_{0i}$) and map ($\nu_{0j}$), and a random slope ($\mu_{1i}$) for urgency (the sole within-subjects experimental manipulation) by participant. Upon seeing that the distributions of the plan time and trace time metrics did not approximate normal distributions, we modified our planned analyses (see Assumption Check).

$$Y_{ij} = \beta_0 + \beta_1 X_{ij} + \beta_2 X_{ij} + \beta_3 X_{ij} + \beta_1\beta_2 X_{ij} + \beta_1\beta_3 X_{ij} + \beta_2\beta_3 X_{ij} + \beta_1\beta_2\beta_3 X_{ij} + \mu_{0i} + \nu_{0j} + \mu_{1i} X_{1j} + \varepsilon_{ij} \quad (1)$$

Time pressure manipulations were fixed-effect predictors, simple contrast-coded [-0.5, 0.5], with the positive value assigned to the expected higher time pressure condition for each factor (less total time, longer waiting times, urgent trials). Treatment coding was preregistered and changed upon learning that deviation or simple contrast-coding was more appropriate for models with interactions. When analyzing the number of trials completed (a participant-level outcome), we used only the two between subjects fixed effects in a single-level regression model

(i.e., without the random intercept for map or the random slope for urgency by participant). We evaluated statistical significance of fixed-effect parameter estimates with p-values generated from two-tailed tests, using Satterthwaite approximations (α = .05).

To examine individual differences in each trial-level outcome, we calculated intraclass correlation coefficients (ICCs). One ICC was calculated as a proportion of response variability associated with participants and another as the proportion of variability associated with the effect of urgency across participants. To examine differences in trial-level outcomes associated with stimuli, we calculated an ICC as the proportion of variability associated with maps. Not specified in the preregistration, we person- and grand-mean centered a trial-level subjective stress variable in H4 and H5 and grand-mean centered all individual difference predictors.

## Results

### Assumption check

Prior to conducting the manipulation check using subjective time pressure scores, a Levene's test with the car package [41] indicated that the variances were sufficiently homogenous across the eight cells in the fixed-effect design ($F(7,444) = 1.97$, $p = .058$). All primary dependent variables except planning time somewhat resembled Gaussian distributions (i.e., subjective time pressure, subjective stress, route length, and number of trials). A generalized linear mixed-effects model with a gamma distribution for the outcome and a log link was used for the planning time outcome (a modification to the preregistration). We transformed those model estimates with the exponential function to report odds ratios. For all models, the distributions of regression residuals were peaked (highly kurtotic).

**Manipulation check: Effects of experimental manipulations on subjective time pressure.** We used two scores for each participant to determine whether the time pressure manipulations increased subjective time pressure. The model used two between-subject factors (total time and wait time) and one within-subjects factor (urgency messaging). The manipulation check (Fig 3) suggested that the within-subject urgency messaging manipulation increased subjective time pressure, $F(1,222) = 227.21$, $p < .001$. There was no evidence that the total time, $F(1,222) = 1.47$, $p = .226$, and wait time, $F(1,222) = 0.22$, $p = .638$, manipulations affected subjective time pressure, and there were no significant interactions ($ps > .138$).

To quantify the conditional probability that the between-subjects manipulations did not affect felt time pressure, we conducted Bayesian hypothesis tests with the BFpack package [42] based on regression models of subjective time pressure during the whole task, and subjective time pressure during each type of trial (urgent and not urgent). We included the between-subjects conditions and their interaction as simple contrast-coded predictors in the first model, and between- and within-subjects conditions and interactions in the second model. The posterior probabilities of null effects using generalized adjusted fractional Bayes factors were high in both models: total time (P0 = .88, P0 = .86), wait time (P0 = .89, P0 = .83), and the interaction (P0 = .78, P0 = .82). We conducted the planned analyses, although the effects of total time and waiting time were not considered linked to subjective time pressure.

### Confirmatory analyses

Table 2 shows descriptive statistics and zero-order correlations for primary variables of interest. Tables 3 and 4 present model results that were interpreted to test the confirmatory hypotheses.

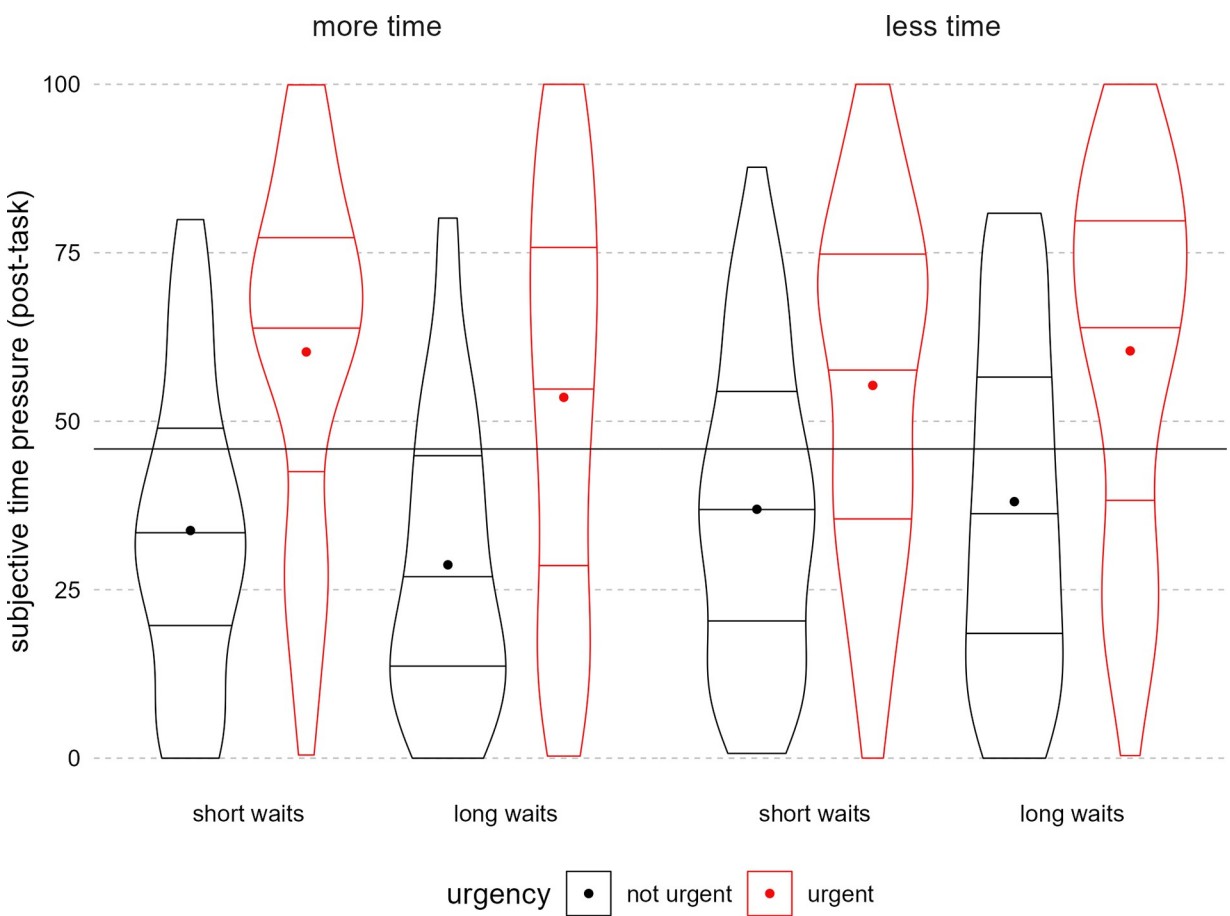

**Fig 3. Post-task subjective time pressure by experimental condition.** Figure displays the density of participant-level data, attenuated at the real data range. The leftmost plotted condition was hypothesized to have the lowest time pressure and the rightmost, the highest. The panels indicate the total time condition, the x-axis the waiting time condition, and the paired plots indicate the urgency condition (red = urgent). In each condition, horizontal lines indicate quartiles. Each dot is the mean for that condition. The overlayed horizontal line is the grand mean across all conditions.

**Table 2. Zero-order correlations between individual-level variables.**

| | M | SD | 1 | 2 | 3 | 4 | 5 | 6 | 7 |
|---|---|---|---|---|---|---|---|---|---|
| 1. Subjective Time Pressure | 50.42 | 26.58 | | | | | | | |
| 2. Trial Subjective Stress | 37.35 | 22.30 | .76** | | | | | | |
| 3. Trial Plan Time | 10.57 | 6.38 | -.04 | .02 | | | | | |
| 4. Trial Tracing Time | 35.93 | 18.73 | .05 | .19** | .55** | | | | |
| 5. Trial Route Length | 1.86 | 0.25 | .11 | .17* | .28** | .56** | | | |
| 6. Number Routes | 30.12 | 16.24 | -.05 | -.23** | -.58** | -.63** | -.30** | | |
| 7. Base Subjective Stress | 33.27 | 24.59 | .51** | .58** | -.08 | -.04 | -.01 | -.06 | |
| 8. Post Subjective Stress | 46.50 | 27.70 | .76** | .88** | -.02 | .14* | .15* | -.13* | .56** |

We computed the mean across trials for each participant for the trial-level outcomes (2, 3, 4, & 5). Base = Baseline; Post = Post-task. **Bold** values are statistically significant, denoted with * for $p < .05$ and

** for $p < .01$.

**H0: Effects of time pressure on the number of routes completed.** We hypothesized that the total time and wait time manipulations would decrease the number of routes planned. Because urgent and non-urgent messaging was randomly assigned on a trial-by-trial basis independent of the total time allocated to complete the task, our preregistered hypothesis that urgent messaging would increase the number of routes compared to non-urgent messaging was ill-conceived. Observing an effect of urgency would have required equal amounts of time allocated to the urgent and not urgent conditions.

Although the total time and wait time manipulations did not increase subjective time pressure or stress, they both led to fewer routes completed (Table 3, Fig 4). Not surprisingly, these results suggest that having more time to do the task increased the quantity of output.

**H1: Effect of time pressure manipulations on trial-level subjective stress.** We hypothesized that having less total time, longer average waiting time, and exposure to urgency messaging would increase subjective stress. On average, participants experienced some subjective stress during the route planning task, rating stress an average of about 37 out of 100 (Table 2).

**Table 3. Experimental models of outcomes at the participant (H0) and trial (H1-H3) levels.**

| Outcome | Routes Completed (H0) | | | | Subjective Stress (H1) | | | | Planning Time (H2) | | | | Tracing Time (H2E) | | | | Route Length (H3) | | | |
|---|---|---|---|---|---|---|---|---|---|---|---|---|---|---|---|---|---|---|---|---|
| *Fixed Effect Predictors* | *b* | *SE* | *t* | *p* | *b* | *SE* | *t* | *p* | *OR* | *SE* | *t* | *p* | *OR* | *SE* | *t* | *p* | *b* | *SE* | *t* | *p* |
| Intercept | 14.61 | 0.46 | 31.78 | < **.001** | 37.40 | 1.5 | 24.30 | < **.001** | 8.73 | 0.43 | 44.07 | < **.001** | 32.57 | 1.96 | 57.99 | < **.001** | 1.859 | 0.054 | 34.21 | < **.001** |
| Less—More Time | -5.87 | 0.92 | -6.38 | < **.001** | 3.88 | 3.0 | 1.30 | .194 | 1.08 | 0.10 | 0.82 | .412 | 1.05 | 0.08 | 0.68 | .495 | 0.013 | 0.029 | 0.46 | .646 |
| Long—Short Waits | -6.19 | 0.92 | -6.73 | < **.001** | -0.49 | 3.0 | -0.16 | .870 | 1.13 | 0.10 | 1.41 | .160 | 1.11 | 0.09 | 1.37 | .171 | 0.008 | 0.029 | 0.29 | .769 |
| Urgent—Not Urgent | -0.57 | 0.35 | -1.62 | .105 | 8.70 | 0.8 | 11.34 | < **.001** | 0.88 | 0.02 | -5.81 | < **.001** | 0.92 | 0.01 | -5.87 | < **.001** | -0.002 | 0.009 | -0.27 | .788 |
| Time × Waits | 1.25 | 1.84 | 0.68 | .497 | 1.91 | 6.0 | 0.32 | .748 | 0.94 | 0.17 | -0.36 | .721 | 1.09 | 0.17 | 0.54 | .590 | 0.064 | 0.057 | 1.12 | .264 |
| Time × Urgency | 0.04 | 0.70 | 0.06 | .955 | 0.01 | 1.5 | 0.01 | .995 | 1.01 | 0.05 | 0.27 | .786 | 1.01 | 0.03 | 0.35 | .727 | 0.011 | 0.018 | 0.60 | .549 |
| Waits × Urgency | -0.22 | 0.70 | -0.31 | .753 | 2.47 | 1.5 | 1.61 | .101 | 0.98 | 0.05 | -0.51 | .613 | 1.00 | 0.03 | 0.09 | .925 | -0.019 | 0.018 | -1.05 | .294 |
| Time × Waits × Urgency | -0.12 | 1.40 | -0.09 | .929 | 3.68 | 3.1 | 1.20 | .231 | 0.98 | 0.09 | -0.27 | .788 | 1.05 | 0.06 | 0.82 | .414 | -0.020 | 0.036 | -0.57 | .566 |
| *Random Effects Clusters* | | | | | | | | | | | | | | | | | | | | |
| $\tau_{00}$ | 40.85 ID | | | | 454.58 ID | | | | 0.14 ID | | | | 0.05 ID | | | | 0.04 ID | | | |
| | | | | | 17.60 map | | | | 0.01 map | | | | 0.02 map | | | | 0.33 map | | | |
| $\tau_{11}$ *urgMsg \| ID* | | | | | 101.85 | | | | 0.05 | | | | 0.02 | | | | 0.00 | | | |
| $\rho_{01}$ | | | | | .06 ID | | | | -.29 ID | | | | -0.28 ID | | | | -.22 ID | | | |
| ICC*ID* | .75 | | | | .61 | | | | .32 | | | | .31 | | | | .09 | | | |
| ICC*urgMsg \| ID* | | | | | .14 | | | | .10 | | | | .09 | | | | .003 | | | |
| ICC*map* | | | | | .02 | | | | .03 | | | | .14 | | | | .69 | | | |
| N | | | | | 120 map | | | | 120 map | | | | 120 map | | | | 120 map | | | |
| | 226 ID | | | | 226 ID | | | | 226 ID | | | | 226 ID | | | | 226 ID | | | |
| Observations | 452 | | | | 6709 | | | | 6757 | | | | 6757 | | | | 6757 | | | |
| Marginal $R^2$ (predictors) | .255 | | | | .033 | | | | .026 | | | | .031 | | | | .001 | | | |
| Conditional $R^2$ | .811 | | | | .748 | | | | .411 | | | | .513 | | | | .782 | | | |

Intercepts for H0, H1, and H3 represent the predicted value of the outcome, and the intercept for H2 is the odds ratio, when all experimental manipulations held constant at the grand average (0). ID specifies the random intercept for participant, map the random intercept for map, and urgMsg | ID is the random slope for urgency by participant. **Bold** *p*-values indicate statistical significance. Tracing time is marked H2E to reflect the exploratory nature of that analysis.

**Table 4. Subjective stress included in models of trial-level outcomes (H4-H5).**

| Outcome | Planning Time (H4) | | | | Tracing Time (H4E) | | | | Route Length (H5) | | | |
|---|---|---|---|---|---|---|---|---|---|---|---|---|
| *Fixed Effect Predictors* | *OR* | *SE* | *t* | *p* | *OR* | *SE* | *t* | *p* | *Estimate* | *SE* | *t* | *p* |
| Intercept | 8.515 | 0.232 | 78.77 | < **.001** | 30.064 | 0.608 | 168.24 | < **.001** | 1.859 | 0.053 | 35.15 | < **.001** |
| Less—More Time | 1.065 | 0.054 | 1.25 | .212 | 1.021 | 0.031 | 0.61 | .496 | 0.007 | 0.028 | 0.25 | .802 |
| Long—Short Waits | 1.115 | 0.056 | 2.17 | **.030** | 1.101 | 0.033 | 3.19 | **.001** | 0.009 | 0.028 | 0.32 | .749 |
| Urgent—Not Urgent | 0.859 | 0.017 | -7.50 | < **.001** | 0.891 | 0.010 | -9.95 | < **.001** | -0.048 | 0.010 | -4.76 | < **.001** |
| Subjective Stress (GMC) | 1.001 | 0.001 | 1.07 | .285 | 1.004 | 0.001 | 6.19 | < **.001** | 0.001 | 0.001 | 2.33 | **.020** |
| Subjective Stress (PMC) | 1.002 | 0.000 | 4.74 | < **.001** | 1.004 | 0.000 | 17.03 | < **.001** | 0.005 | 0.000 | 17.52 | < **.001** |
| Time × Waits | 0.943 | 0.094 | -0.59 | .556 | 1.108 | 0.067 | 1.71 | .088 | 0.060 | 0.056 | 1.06 | .287 |
| Time × Urgency | 1.014 | 0.040 | 0.35 | .728 | 1.012 | 0.023 | 0.53 | .597 | 0.010 | 0.019 | 0.54 | .589 |
| Waits × Urgency | 0.980 | 0.039 | -0.51 | .612 | 0.991 | 0.023 | -0.38 | .705 | -0.031 | 0.019 | -1.59 | .111 |
| Time × Waits × Urgency | 0.993 | 0.079 | -0.09 | .928 | 1.024 | 0.047 | 0.51 | .608 | -0.034 | 0.039 | -0.89 | .375 |
| *Random Effects Clusters* | | | | | | | | | | | | |
| $\tau_{00}$ | 0.14 ID | | | | 0.05 ID | | | | 0.04 ID | | | |
| | 0.01 map | | | | 0.02 mapID | | | | 0.31 map | | | |
| $\tau_{11urgMsg \mid ID}$ | 0.05 | | | | 0.02 | | | | 0.00 | | | |
| $\rho_{01}$ | -.29 ID | | | | -0.28 ID | | | | -.26 ID | | | |
| $ICC_{ID}$ | .32 | | | | .32 | | | | .09 | | | |
| $ICC_{urgMsg \mid ID}$ | .11 | | | | .10 | | | | .01 | | | |
| $ICC_{map}$ | .03 | | | | .13 | | | | .68 | | | |
| N | 120 map | | | | 120 map | | | | 120 map | | | |
| | 226 ID | | | | 226 ID | | | | 226 ID | | | |
| Observations | 6709 | | | | 6709 | | | | 6709 | | | |
| Marginal $R^2$ (predictors) | .027 | | | | .097 | | | | .014 | | | |
| Conditional $R^2$ | .413 | | | | .548 | | | | .784 | | | |

Intercepts for H5 represent the predicted value of the outcome, and the intercept for H4 is the odds ratio, when all experimental manipulations held constant at the grand average (0) and subjective stress in H4-5 held constant at the grand mean and the mean for each participant. ID specifies the random intercept for participant, map the random intercept for map, and urgMsg | ID is the random slope for urgency by participant. GMC is grand mean-centered, which indicates between-persons variability in subjective stress. PMC is person mean-centered, which indicates within-person variability in subjective stress. **Bold** *p*-values indicate statistical significance.

As hypothesized, participants reported higher levels of subjective stress on urgent trials than on non-urgent trials (Fig 5). Contrary to hypotheses, the other manipulations–total time and wait time–did not significantly affect subjective stress. There were no significant interactions (Table 3).

**H2 & H3: Effect of time pressure manipulations on performance.** We hypothesized that having less total time, longer wait times, and a greater sense of urgency each would decrease the time spent planning and increase the route length. On average, participants spent about 10 seconds planning per trial. Consistent with our hypothesis, participants spent less time planning on urgent trials than on non-urgent trials (see Fig 6). However, there were no statistically significant main or interaction effects of the total time or wait time manipulations on planning time.

The routes traced averaged 1.86 arbitrary standardized units. Contrary to our hypothesis, none of the experimental manipulations affected route length (without controlling for subjective stress), and there were no significant interactions (see Fig 7 and Table 3).

**H4 and 5: Associations between trial-level subjective stress and performance.** In addition to examining subjective stress as a trial-level outcome, we also examined subjective stress

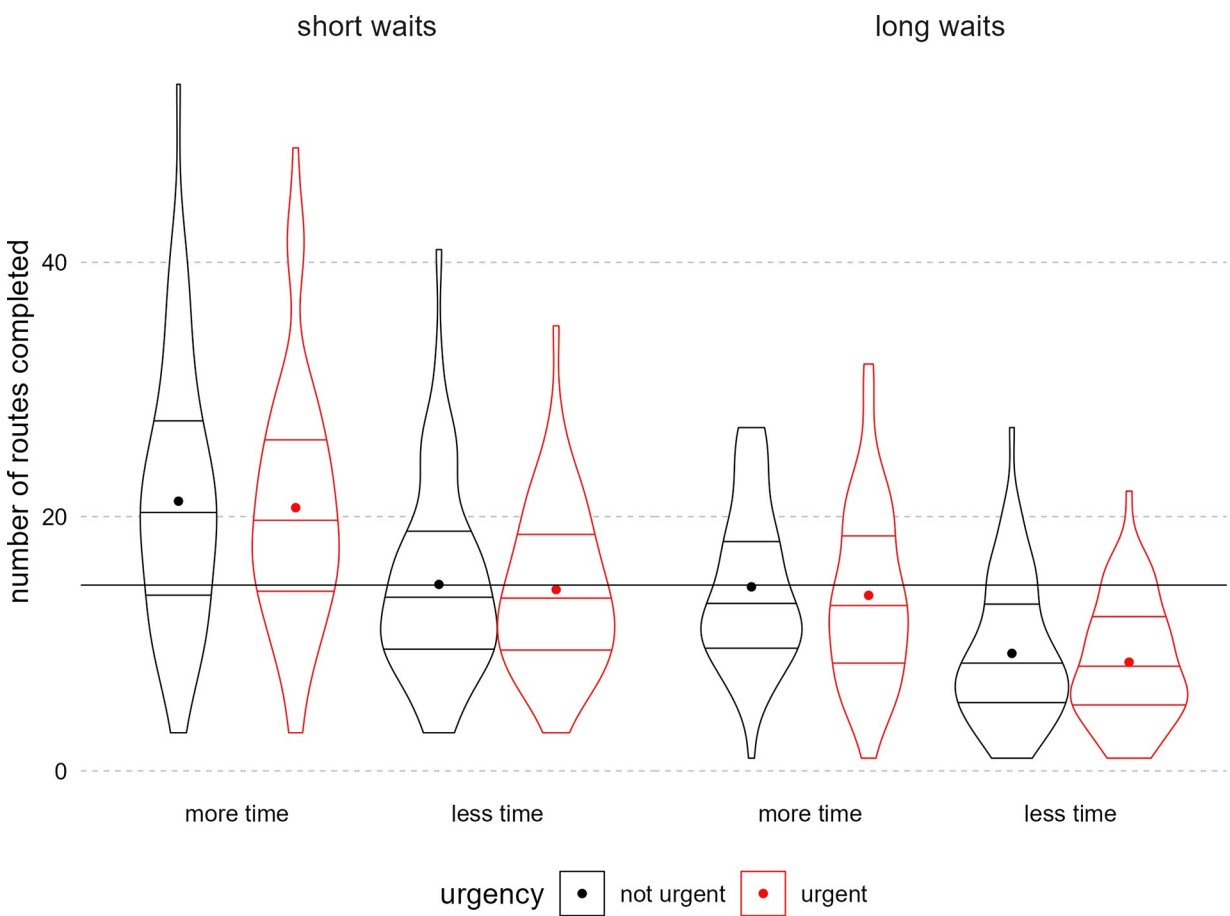

**Fig 4. Number of trials completed by total time and waiting time.** Figure displays density of participant-level data, number of routes completed, attenuated at the real data range. The far-left plotted condition was hypothesized to have the most maps completed and the far-right, the least. The panels indicate the total time condition, the x-axis the waiting time condition, and the paired plots indicate the urgency condition (red = urgent). In each condition violin, horizontal lines indicate quartiles. Each dot is the mean for that condition. The overlayed horizontal line is the grand mean across all conditions.

as a trial-level fixed-effect predictor of performance in two models (H4-5; see Table 4). We hypothesized that subjective stress reported after each trial would be associated with decreased planning times and increased route lengths. We tested this by including grand-mean and person-mean centered subjective stress predictors in the model. As hypothesized, between- and within-subjects variation in subjective stress was associated with increased route lengths when all other factors in the model were held at an average. However, contrary to our hypothesis, within-subjects variation in subjective stress was also associated with increased planning times (when all other factors in the model were held at an average). When considering the magnitude of these associations, the effect of subjective stress reflects a 1-unit change on a scale from 0 to 100. Interestingly, when subjective stress was held constant in the models for H4 and H5, there was a statistically significant effect of waiting periods on increased plan time.

**H6: Individual differences in wayfinding performance outcomes.** We hypothesized that there would be reliable individual differences in the trial-level outcomes. To assess individual differences, we examined the ICCs corresponding to the random intercept for participant and the random slope for urgency by participant (Table 3). Around 60% of the variability in trial-level subjective stress, 30% of the variability in planning time, and 10% of the variability in

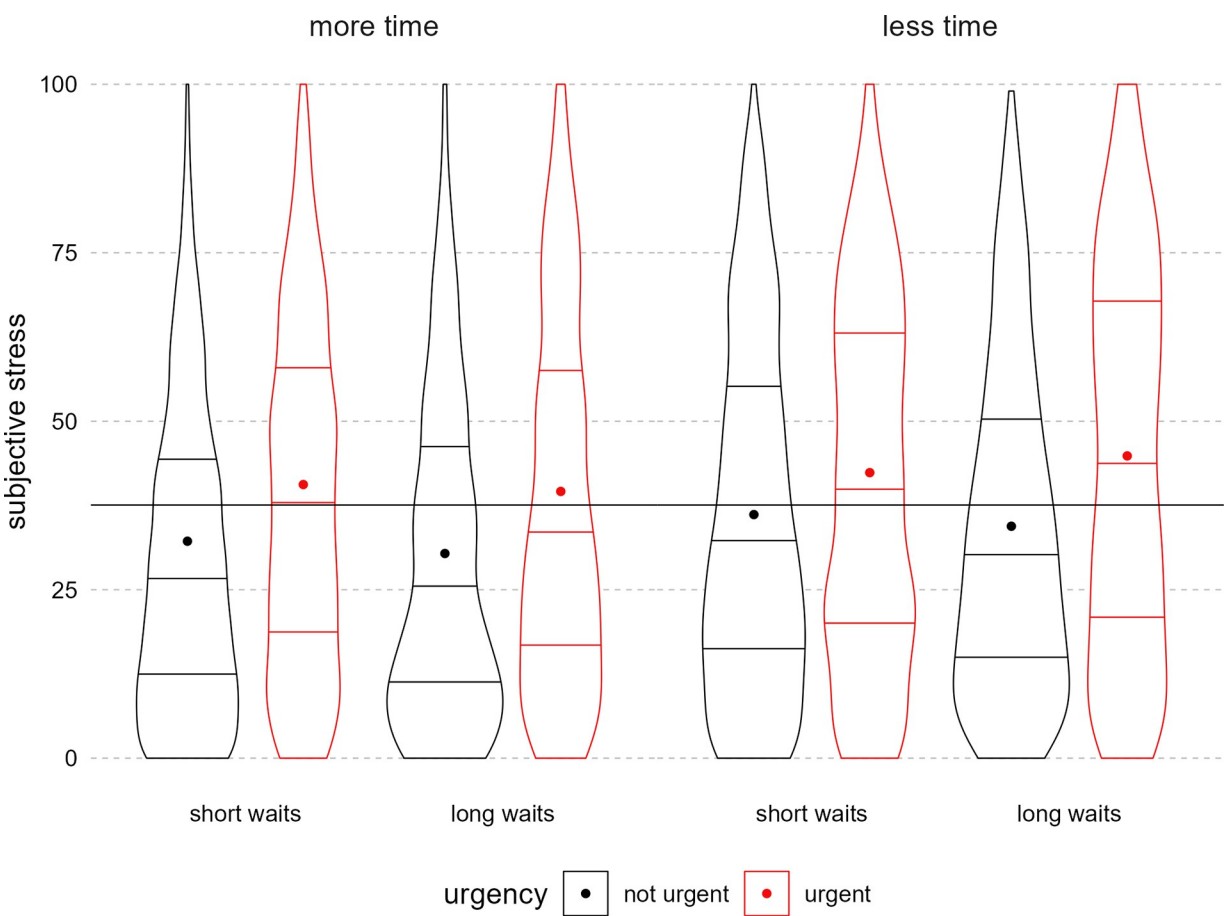

**Fig 5. Subjective stress during the task by total time, waiting time, and urgency messaging.** Figure displays trial-level subjective stress. The far-left plotted condition was hypothesized to have the lowest subjective stress and the far-right, the highest. The panels indicate the total time condition, the x-axis the waiting time condition, and the paired plots indicate the urgency condition (red = urgent). In each condition violin, horizontal lines indicate quartiles. Each dot is the mean for that condition. The overlayed horizontal line is the grand mean across all conditions.

route length could be explained by individual differences. Smaller portions (14%, 11%, 0.3%, respectively) of variability were explained by the effect of urgency across participants (i.e., the random slope). Map stimuli explained very little of the variability in subjective stress (about 2%) and planning time (3%), but about 70% of the variability in route length. H6 was labelled H7 in preregistration.

## Exploratory analyses

We examined the time spent tracing each route as an exploratory outcome (Fig 8). Tracing time appeared to have a similar distribution to planning time, so we fit a generalized linear mixed-effects model with a gamma distribution and a log link (Table 3). Participants averaged approximately 30 seconds per route. Urgency messaging reduced tracing time. When controlling for subjective stress at the individual and trial level (Table 4), urgency still reduced tracing time while long waiting periods increased tracing time. Both subjective stress measurements were associated with increased tracing time.

We also sought to better understand participants' experience of stress and how it relates to performance during the task by examining three qualitatively different subjective stress states

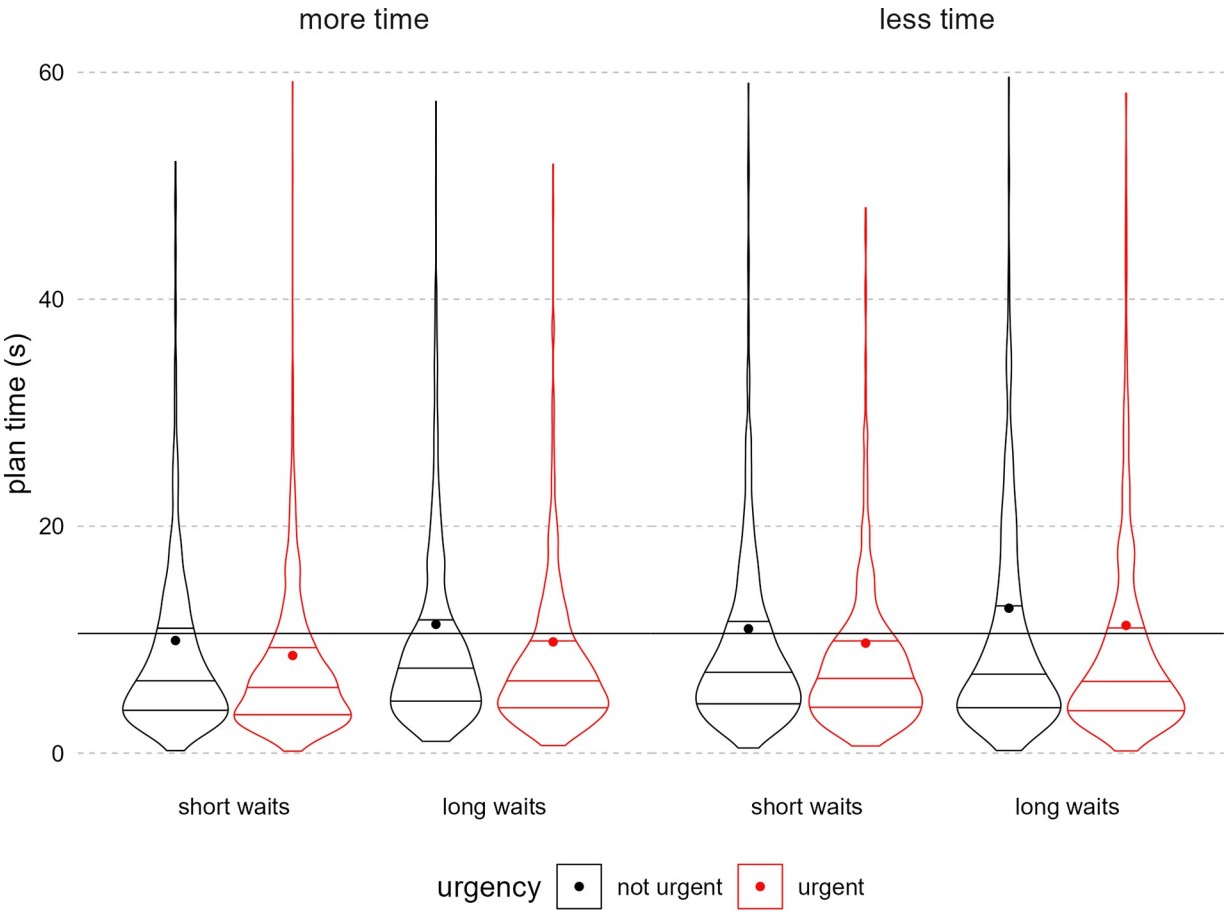

**Fig 6. Planning time by total time, waiting time, and urgency messaging.** Figure displays density of trial-level planning time in seconds, attenuated at the real data range. The far-left plotted condition was hypothesized to have the shortest planning time and the far-right, the longest. The panels indicate the total time condition, the x-axis the waiting time condition, and the paired plots indicate the urgency condition (red = urgent). In each condition violin, horizontal lines indicate quartiles. Each dot is the mean for that condition. The overlayed horizontal line is the grand mean across all conditions.

at baseline and at post-task (S1 Fig); the method, results, and discussion are reported in the S1 Text. We further probed differences in map type and ideal route length. Importantly, none of these analyses fundamentally alter conclusions drawn from the confirmatory analyses.

We conducted additional exploratory models to test differences in performance due to mouse ($n$ = 189) or trackpad ($n$ = 95) use. Specifically, we added a factor coded for device use as a covariate to the models described in H0-H5. We did not consider interactions between device type and the experimental manipulations. In exploratory models of the two performance outcomes, when controlling for subjective stress, trackpad use was associated with shorter planning time and shorter routes, but including device in the models did not alter the conclusions of confirmatory hypothesis testing.

## Discussion

Maule and Hockey [43], referring to the idea of manipulating time pressure by imposing different deadlines, stated that "unless we can develop a more comprehensive account. . . it will not be possible to determine whether this way of operationalizing time pressure is appropriate"

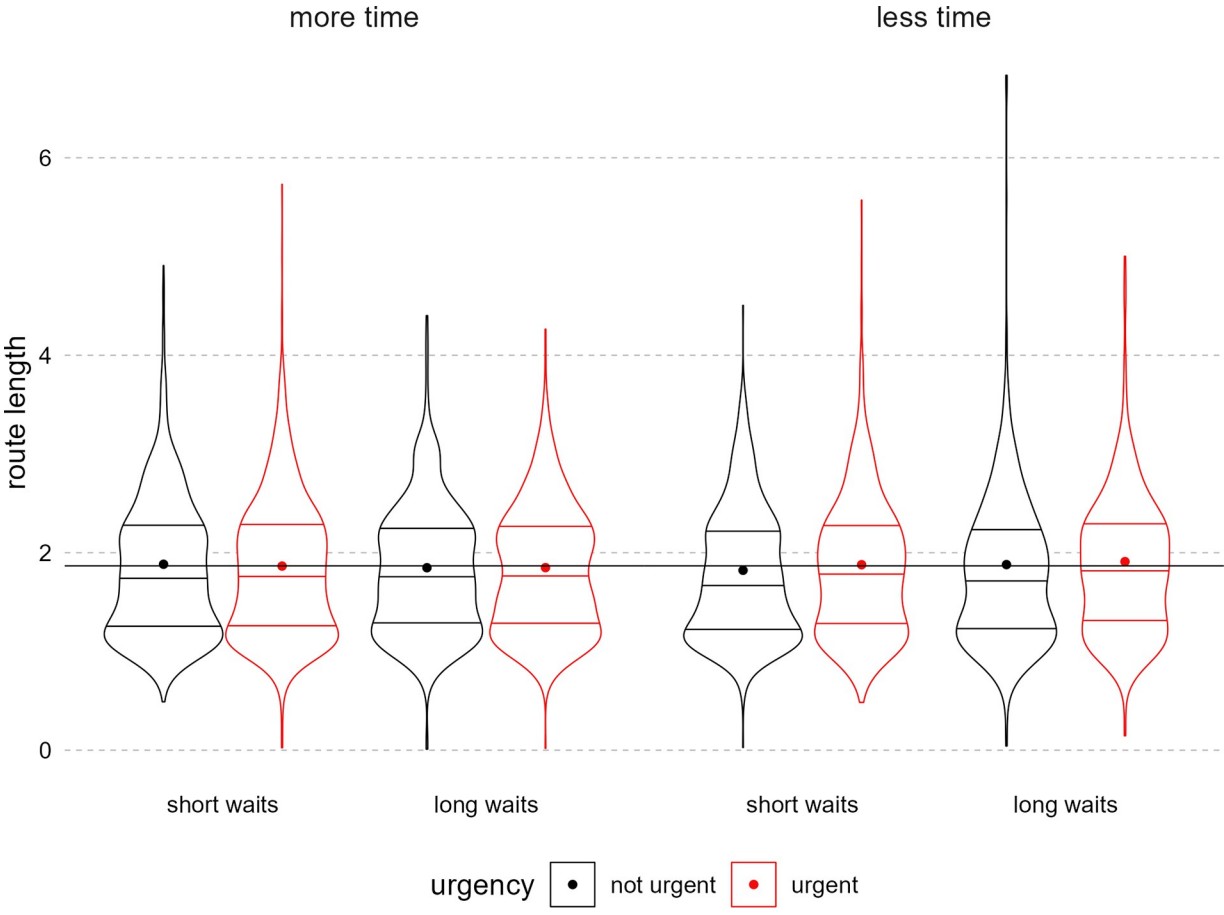

**Fig 7. Route length by total time, waiting time, and urgency messaging.** Figure displays density of trial-level route length, attenuated at the real data range. The far-left plotted condition was hypothesized to have the shortest route length and the far-right, the longest. The panels indicate the total time condition, the x-axis the waiting time condition, and the paired plots indicate the urgency condition (red = urgent). In each condition violin, horizontal lines indicate quartiles. Each dot is the mean for that condition. The overlayed horizontal line is the grand mean across all conditions.

(p. 85). Current work focused on perceived time pressure considers factors other than explicitly changing deadlines [17]. Our study took a comprehensive approach to understanding effects of time pressure with three different experimental operationalizations on subjective stress and route planning.

Time pressure is unavoidable when navigating the world and the feeling of being stressed with navigating is not just supported by research, but with the lived experience of many people. In this study, we introduced a new, open-source method to study route planning in a controlled laboratory-style task and attempted to manipulate time pressure by giving people less time to complete a task, making them wait, and sending messages of urgency. We were primarily trying to determine if time pressure manipulations affected subjective stress and route planning performance, if subjective stress was associated with performance outcomes, and whether there were individual differences in subjective stress and performance outcomes.

In general, our task provoked some subjective time pressure and stress (more than a third of the way up a continuous scale from 0 to 100, on average). Unsurprisingly, participants completed fewer routes when they had less total time or longer waiting times (i.e., less effective amount of time available to complete the task). On the other hand, there were no substantive

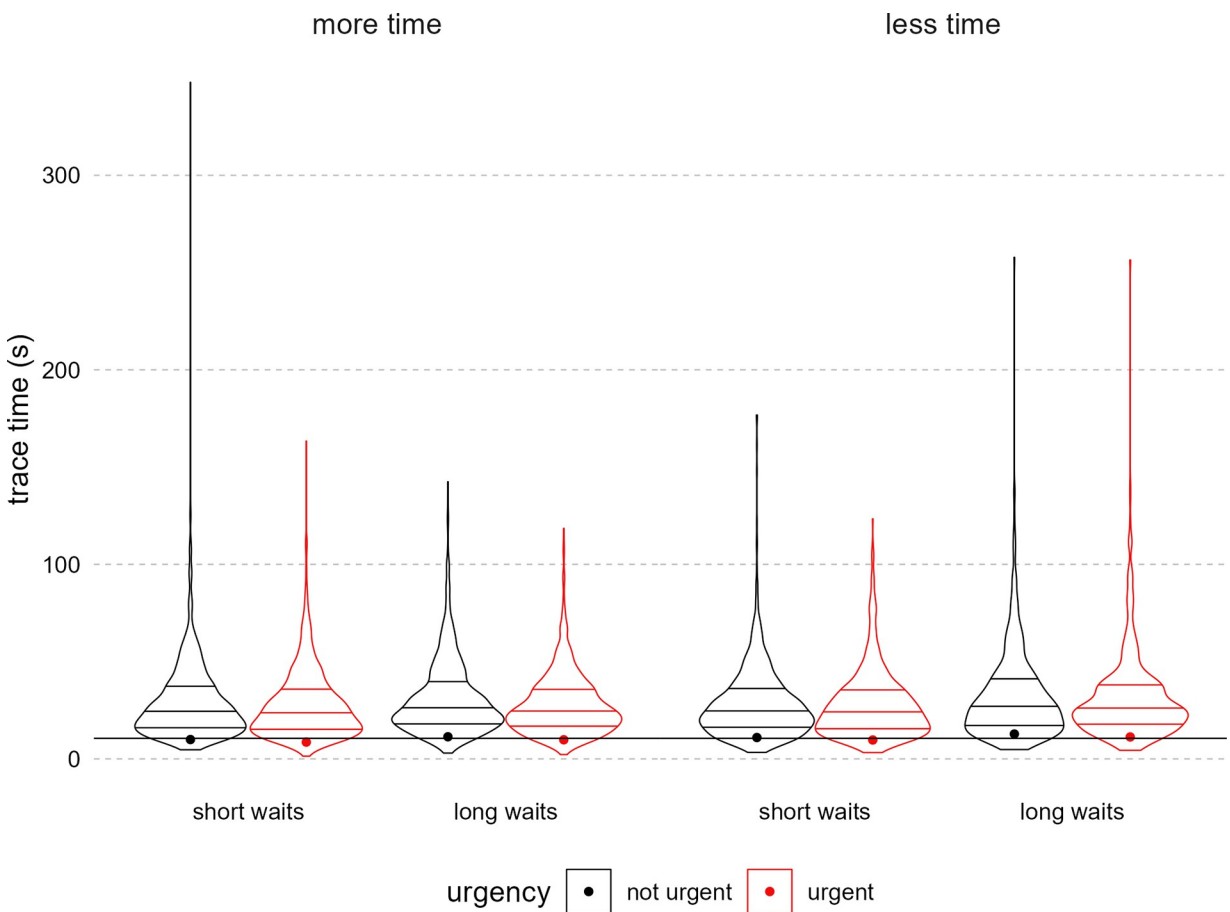

**Fig 8. Planning time by total time, waiting time, and urgency messaging.** Figure displays density of trial-level tracing time in seconds, attenuated at the real data range. The panels indicate the total time condition, the x-axis the waiting time condition, and the paired plots indicate the urgency condition (red = urgent). In each condition violin, horizontal lines indicate quartiles. Each dot is the mean for that condition. The overlayed horizontal line is the grand mean across all conditions.

performance differences in 35 minutes vs. 25 minutes in terms of subjective time pressure, subjective stress, or trial-level performance metrics. Similarly, encountering longer waiting periods during the task had little effect on subjective time pressure, subjective stress, or trial-level performance metrics (with the exception that, at average levels of subjective stress, waiting periods led to longer planning and tracing time). Thus, ironically, the two manipulations that affected the amount of time participants had to do the task did not function as temporal stressors in this study (i.e., a condition which leads to both time pressure and subjective stress). Indeed, there were no significant effects of total time or waiting time on subjective time pressure. We inferred from these data, as others have stated anecdotally [25, 32, 44, 45], that altering the actual amount of time allocated to complete a task alone does not necessarily induce experiences of time pressure or stress.

In contrast to manipulations of total time and waiting time, imposing a sense of urgency by telling participants to "*Hurry up!*" with a stopwatch style count-up timer increased subjective experiences of time pressure and stress. As such, urgency messaging did function as a temporal stressor. Imposing a sense of urgency also affected wayfinding performance, but not exactly as hypothesized. We expected that time pressure manipulations would reduce planning time but increase route length. We also expected an overall negative correlation between planning time

and route length, potentially indicating that taking more time to plan would lead to better performance (i.e., shorter, hence more efficient, routes). Instead, imposing a sense of urgency affected both outcomes in the same direction, and zero-order correlations showed a moderate positive association between planning time and route length (Table 2). Therefore, our idea that taking more time to plan would improve performance was not supported.

Additionally, we thought that the time pressure manipulations and subjective stress measured during the task would be associated with performance in similar ways. However, whereas urgency-induced time pressure *decreased* planning time and led to shorter routes, subjective stress measured after each trial was associated with *increased* planning time and longer routes. We do not know why these associations appeared to be opposite, but one possibility is that subjective stress was influenced by appraisals of performance; spending less time planning and tracing shorter routes may have made participants feel less stressed afterward given the constraints on time. It is also possible that experiencing greater stress after being told to hurry up may have made participants less efficient (i.e., slowed them down, prompted longer routes). In that case, subjective stress is a mechanism by which urgency messaging affects performance. Future research examining the possibility of indirect effects of urgency messaging on performance outcomes is warranted.

## Strengths, limitations, & future directions

We created a new computerized procedure to measure route planning and compared the effects of three time pressure manipulations on subjective stress and performance outcomes. We also established the presence of meaningful individual differences in these outcomes. We took a rigorous approach to understanding time pressure and measuring performance, but given the novelty of the route planning task, we do not know whether the performance metrics are associated with real-world navigation.

This study used a representative sample of people in the U.S. with Prolific accounts by age, gender, and ethnicity. The results of this study may therefore be generalized to English-speaking adults (with a Prolific account) who are residing in the United States. However, if one were interested in a specific subgroup, non-English speakers, or those living outside the United States, additional research would help confirm generalizability. For example, we did not focus explicitly on groups such as the military or first responders, who often need to plan routes in novel environments while under time pressure, for whom this research may be particularly relevant.

Furthermore, we did not consider route planning with restricted-information. Restricted information refers to situations when the entire environment is not visible in at least some part of the trip, such as when wayfinding in real world environments or virtual 3-D environments without a map. The construct of interest in this task is route planning, which is under the umbrella of spatial navigation but not necessarily real-world wayfinding. We also minimized the influence of learning and memory in this operationalization of route planning in a new environment.

We managed to induce a moderate amount of time pressure (about halfway up a scale from "no pressure" to "extreme pressure"). Average subjective stress was about 13 points lower than average time pressure, and not all participants found the task stressful. Stronger time pressure or greater incentives could help reveal the full spectrum of time pressure effects on performance outcomes in our task.

We were surprised by the saliency of urgency messaging, as well as the absence of main effects of total time and waiting time on subjective stress and performance outcomes. It is possible that the differences in total time and wait time were not large enough to discern signal from noise in our between-subjects manipulations. By contrast, the within-subjects trial-level

manipulation of temporal stress via urgency messaging is a good template for future research. Future research could consider effects of total time and waiting times that are manipulated at the trial level within subjects.

Another direction future research is to distinguish between effects of stress and urgency. For example, a threat of shock procedure [e.g., 46] could be crossed with urgency messaging. Furthermore, continuous measures of physiological stress could be collected during performance. Navigation research in a driving simulator demonstrated that adaptive physiological stress responses (e.g., increased pupil diameter, heart rate) co-occurred with negative experiential responses (e.g., feeling of hurry, frustration) when time pressure was imposed by temporal constraints and a virtual passenger [27]. These short-term physiological indicators could be used to measure an ongoing stress response during route planning task.

## Conclusions

For a long time, people have been interested in the consequences of time pressure. Researchers manipulate time pressure in different but typically singular ways, such as giving people less time to do a task, imposing waiting periods, or instilling a sense of urgency–but not all three separately. As a result, it is unknown to what extent these three manipulations of temporal pressure a) contribute to the sense of subjective stress, and b) impact task performance. Our work addressed this in the context of a novel wayfinding task that involved planning and tracing routes on maps.

Ultimately, we found that only a sense of urgency prompted increases in time pressure and subjective stress, making it qualify as a temporal stressor. This temporal stressor affected performance. Although urgency increased subjective stress, urgency messaging and trial-level subjective stress had opposite associations with performance outcomes. These unexpected findings set the stage for replication studies and further process-focused tests. More research would be necessary to determine whether these task-based measures can predict real-world wayfinding performance.

### Constraint on generality

Our route planning task design was constrained by the goal to test effects of time pressure on spatial performance online. With consideration for financial and practical resources, and the knowledge that we would conduct online research, we aimed to generalize results to adults in the United States. We selected Prolific as the recruitment platform before selecting a sample size. We had to balance hourly compensation and sample size within the budget. We also paid Prolific extra to obtain a representative sample in the United States. This representative sampling employed stratified (quota) sampling based on US census data from 2010, using three criteria: age (five 9-year brackets: 18–27, 28–37, 38–47, 48–57, and 58+), sex (male and female), and ethnicity (Asian, Black, Mixed, Other, and White). However, it is possible that even with a representative sample, Prolific participants may be different than the general population. For example, results from Prolific participants who are older may not generalize as well as for those who are younger [47].

## Supporting information

**S1 Text. Route review procedures and exploratory analyses.**
(DOCX)

**S1 Fig. Engagement, distress, and worry at baseline and post-task.** Figure displays participant-level data. The legend labels, from top to bottom, correspond with the violin plots from left to right. For each violin (i.e., combination of type of subjective stress state and

measurement time), the horizontal lines indicate quartiles and the dot indicates the mean.
(TIF)

**S1 Data. Map coordinates.** Coordinates in decimal degrees for all map stimuli.
(CSV)

## Acknowledgments

Research assistants Delaney Clarke, Marleigh Giliberto, Florence Grenon, Xinyi (Selena) Hu, Anna Kendall, Katrina Yuzefpolsky, and JoJo Zhang reviewed the performance data. XH and JZ contributed to programming and management during the performance data review.

## Author Contributions

**Conceptualization:** Paul E. Plonski, Prsni Patel, Kathryn L. Ossenfort, Holly A. Taylor, Heather L. Urry.

**Data curation:** Paul E. Plonski.

**Formal analysis:** Paul E. Plonski, Heather L. Urry.

**Funding acquisition:** Holly A. Taylor, Heather L. Urry.

**Investigation:** Paul E. Plonski.

**Methodology:** Paul E. Plonski, Prsni Patel.

**Project administration:** Paul E. Plonski.

**Software:** Paul E. Plonski.

**Supervision:** Heather L. Urry.

**Validation:** Paul E. Plonski, Heather L. Urry.

**Visualization:** Paul E. Plonski.

**Writing – original draft:** Paul E. Plonski.

**Writing – review & editing:** Paul E. Plonski, Holly A. Taylor, Tad T. Brunyé, Heather L. Urry.

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
