## [Decision Letter · Decision Letter 0]

6 Mar 2024

PONE-D-23-34568How much time to get where? Wayfinding and subjective stress under time pressurePLOS ONE

Dear Dr. Plonski,

Thank you for submitting your manuscript to PLOS ONE. After careful consideration, we feel that it has merit but does not fully meet PLOS ONE’s publication criteria as it currently stands. Therefore, we invite you to submit a revised version of the manuscript that addresses the points raised during the review process.

We look forward to receiving your revised manuscript.

Kind regards,

Ioanna Markostamou, Ph.D.

Academic Editor

PLOS ONE

Journal Requirements:

3. Please note that funding information should not appear in the Acknowledgments section or other areas of your manuscript. We will only publish funding information present in the Funding Statement section of the online submission form. Please remove any funding-related text from the manuscript. 

4. Please ensure that you refer to Figure 7 in your text as, if accepted, production will need this reference to link the reader to the figure.

5. Please upload a new copy of Figures 1-3 as the detail is not clear. Please follow the link for more information: 

https://blogs.plos.org/plos/2019/06/looking-good-tips-for-creating-your-plos-figures-graphics/

https://blogs.plos.org/plos/2019/06/looking-good-tips-for-creating-your-plos-figures-graphics/

6. We note that Figures 1-3 in your submission contain map images which may be copyrighted. All PLOS content is published under the Creative Commons Attribution License (CC BY 4.0), which means that the manuscript, images, and Supporting Information files will be freely available online, and any third party is permitted to access, download, copy, distribute, and use these materials in any way, even commercially, with proper attribution. For these reasons, we cannot publish previously copyrighted maps or satellite images created using proprietary data, such as Google software (Google Maps, Street View, and Earth). For more information, see our copyright guidelines: http://journals.plos.org/plosone/s/licenses-and-copyright.

1) You may seek permission from the original copyright holder of Figures 1-3 to publish the content specifically under the CC BY 4.0 license.  

2) If you are unable to obtain permission from the original copyright holder to publish these figures under the CC BY 4.0 license or if the copyright holder’s requirements are incompatible with the CC BY 4.0 license, please either i) remove the figure or ii) supply a replacement figure that complies with the CC BY 4.0 license. Please check copyright information on all replacement figures and update the figure caption with source information. If applicable, please specify in the figure caption text when a figure is similar but not identical to the original image and is therefore for illustrative purposes only.

**Additional Editor Comments:**

Your paper was sent for review to two highly knowledgeable experts in the subject area you are investigating. I have now received their comments back and have read through your paper carefully myself. Please find the two reviews of your manuscript at the end of this letter.

As you will see when you read their critiques, the reviewers pointed out that the paper is addressing an interesting topic. However, both reviewers raised some major concerns that need to be fully addressed in order for the paper to be suitable for publication. The most important concerns were about the suitability of the operationalisation of the study’s constructs, especially with regards to the experimental manipulations employed and the task’s internal and ecological validity. Reviewers also noted that several important methodological details were not explained in sufficient detail and called for more information to be provided when outlining the design, materials, and outcome measures of the study. Moreover, both reviewers felt they struggled to understand the tables and figures of the manuscript and made some critical suggestions to improve the Introduction and Discussion sections.  

I believe that addressing all the points raised by the reviewers would substantially improve your manuscript and increase its potential impact. Therefore, I would like to offer the opportunity to submit a revised version of the manuscript that addresses all the points raised by the reviewers.

Reviewers' comments:

Reviewer's Responses to Questions

**Comments to the Author**

1. Is the manuscript technically sound, and do the data support the conclusions?

Reviewer #1: Partly

Reviewer #2: Partly

2. Has the statistical analysis been performed appropriately and rigorously? 

Reviewer #1: Yes

Reviewer #2: Yes

3. Have the authors made all data underlying the findings in their manuscript fully available?

Reviewer #1: Yes

Reviewer #2: Yes

4. Is the manuscript presented in an intelligible fashion and written in standard English?

Reviewer #1: Yes

Reviewer #2: Yes

5. Review Comments to the Author

**Reviewer #1: **

This is a well written paper which addresses and interesting question and render some interesting results. Having said this, I have a couple of major concerns which I will outline below. I believe that these concerns need to be addressed before the paper can be published.

Major concerns:

The authors state that the urgency manipulation affect planning time and planning performance. At this point, I have a couple of doubts: My main concern is that participants do not need to have a complete plan at the end of the planning phase, but that they feel urged by the instructions to end the planning phase without having a fully formed plan. Instead they continue planning or refining their route plan while they are tracing the route. So, an alternative explanation of their findings is that participants only form coarse plans under time pressure, rather than sub-optimal plans, which they refine during the tracing phase. This might also explain why routes under time pressure instructions were less efficient than without explicit instructions. I am not convinced that the authors can address this point unless they run additional experiments, as the current procedure does not allow them to assess how refined route plans are at the end of the planning phase.

Description of the task and materials: The authors should report in detail how the map stimuli were designed, what criteria they used to design the maps, how many alternative routes between start and destination were (reasonably) available, etc. The authors also mention that some map stimuli had additional waypoints. Depending on where the waypoints were positioned, they could either reduce or increase the difficulty of the planning task and affect the number of reasonable alternatives participants need to consider. Overall, the description of the materials and methods should be improved. For example, the authors need to explain how participants were instructed - were they instructed to plan the shortest possible route?

Detailed comments:

The authors refer to their task as a wayfinding task. I believe that the community would expect wayfinding tasks to have an element of actual navigation/movement through space, rather than just planning routes from maps. I would therefore suggest that the authors think about the terminology they use as the address one aspect of wayfinding with maps, i.e. the initial planning stage from maps, but their work cannot speak to how stress might affect the execution of the plan (which is more than just locomotion as it also required memory and decision making to actually follow the route). This is further supported by Lynch’s quote “the process of determining and following a path

or route between an origin and a destination”.

I find it difficult to reconcile H0 and H2 which seem to contradict each other: H0 states that less time would result in fewer trials, while H2 states that less time should result in quicker planning, i.e. shorter trials, which should increase the number of trials. This should be clarified. I assume this has to do with substantially longer waiting times than planning times, however, this would then render H0 trivial.

I am not sure if this relates to a requirement of the journal in which case I would suggest to redirect this comment to the editor, but I much prefer having figures embedded in the text. This makes it so much easier for reviewers to read and assess the paper as there is no need to endlessly scroll back and forth through the document.

Do you have any information about the interaction device participants used to complete the task. I would imagine that completing the task with a mouse or trackpad could substantially affect the times required to trace a route.

I am unclear what the following text means: “A map stimulus was presented at the start of each trial. The cursor appeared as a blue dot when ‘s’ was pressed to signal the end of the planning phase. The screen above was after a route was traced, but before ‘f’ was pressed to signal that the route was

complete.”

Fig 3 suggests that it may take some time to find the start and destination location which were only two symbols among a lot of additional markers in the maps and it is unclear what a waypoint is (referred to in the text but not explained in detail). The screenshots of the maps stimuli are of low image quality which makes it difficult to read the maps and therefore understand the task. Was the time to find the start location, the destination and potential waypoints included in the planning phase?

The authors state that the planning phase is distinct from the tracing phase. What prevents participants from moving into the tracing phase without having completed the plan and using time in the tracing phase to complete the planning process (i.e. while planning)? I believe this is an important issue to address as some route planning literature suggests that routes unfold during navigation (even though participants are not actually navigating in this experiment). In any case, I could imagine that participant who are put under time pressure might be tempted to start tracing the route without having a complete plan. Instead they could start tracing roughly in the direction of the destination and completing the plan as they trace (which might also explain longer trajectories under time pressure).

Figure caption seem to be missing or they are very short. I would have found it helpful to have more informative figure captions explaining the figures in more detail. For example, Fig3 depicts numbers above the map stimuli and the meaning f these numbers were first explained in the Procedure section whereas I believe it should be presented where the materials are presented and in the Figure caption.

“The first between-subjects manipulation was a countdown timer ...” -> I think it would be more precise to say that the time was the manipulation, as both conditions had a timer (if I understand correctly).

A description of the map stimuli or the route planning tasks is missing. How were these maps designed, what were the criteria for designing the route planning tasks? How were waypoints introduced. How many alternative paths were possible, how were the maps selected for each participants, how did the authors ensure that participants in the different conditions had comparable planning tasks, etc. All this is crucial information which seems to be missing from the manuscript and which needs to be explained in detail.

“The routes traced averaged 1.86 “height units””. It is unclear what that means. The authors should explain exactly how they calculated the route length, how the shortest possible route was determined and how they calculated participant performance. Route planning performance should ideally be described as a percentage above optimal or using a similar measure.

What exactly were the instructions that participants received. Were they instructed to plan the shortest possible route? Do the authors think that the requirement to trace the routes could impact participants planning strategies. Could they, for example, prefer longer routes that are easier to trace, as they have fewer turns? This should be addressed in the revision.

Figure 7 shows that mean planning time is far higher than median planning time suggesting that outliers strongly affect the data. Have the authors conducted outlier analyses? If not, I would suggest to do so and to rerun the analyses.

How did planning time differ for trials with and without waypoints?

The strength and limitations and the future research sections in the discussion are almost half of the discussion. I believe this is too long and is something I would expect in a thesis rather than a journal publication. I would suggest to shorten these sections.

**Reviewer #2: **

This paper examined the contributions of three aspects of time pressure, total amount of time allocated to complete a task, the length of wait times interspersed between trials, and instructions to hurry up. Results indicated that only instructions to hurry up (urgency) increased subjective ratings of stress at the trial level. The results were somewhat contradictory in that urgency decreased planning time and route length and also increased subjective stress, but subjective stress was associated with both increased planning time and longer routes. The authors speculate that subjective stress may have reflected participants self-assessments of their performance, rather than the stress they felt at the beginning of the trial when told to hurry up.

The goal of examining the effects of different aspects of time pressure on navigation is a good one, this is a solid, well conducted experiment, and the results are appropriately analyzed and interpreted.

However I have several reactions that I feel could be addressed a revision to improve the contribution of this paper. Specifically, I am concerned that two of the manipulations would not have been expected, a priori, to impose time pressure (at least if I am understanding them correctly), so it is not surprising that they did not, and I also question the ecological validity of the task. In addition, I had some questions about the methodology and it was hard to integrate the text and figures.. Finally the general discussion seems overly concerned with the limitation of the study and future directions, rather than arguing for the contributions of this research.

First, it is not clear to me how the total time manipulation is a time pressure manipulation at all, as it seems that what was varied was the total amount of time that people had to do a number of route planning trials, but there was no indication of how many trials they had to do in this time, and it seems that at a trial level they were told to take their time. Imposing a time limit per trial would have been a better manipulation (and maybe this was done but that is not what I understood from what I read)

Second, I could not find any information on whether the wait times were within trials or between trials. I think they would be more likely to induce stress if they were within trials, so if they were between trials again I would not have thought, a priori, that this would have induced time pressure or stress.

Third, the task of planning and then tracing a route on a map does not seem very ecologically valid, as it does not include the locomotion aspect of wayfinding, that is, the experience of navigating under time pressure in the real world, something that the authors themselves admit in the general discussion.

Fourth, as noted above, I had work hard to understand the tables and figures, as the variable and column names in the figure (e.g.. low/high, low/low) were not explained in the captions (I figured them out but had to work at it) and the figures were hard to understand because all figures were at the end of the manuscript, while their figure captions appeared in the manuscript itself, and the figures at the end were not labelled by number.

The authors motivation could also be strengthened. It seemed to come down to “nobody has done this before”, but not every study that has not been done should be done, so that is not sufficient justification. Specifically, he introduction could be improved by some more theoretical or at least practical motivation for why different aspects of time pressure might have different effects, to motivate why this research is important. Similarly much of the general discussion was devoted to a list of limitations of the present study, as if the authors were writing a negative review of their own paper. For example the broad sample used is a strength of the paper (much better than the typical college sample) but the authors seem to almost interpret it a limitation by questioning whether it truly represents the population. Similarly the section on future possible studies seems too long (this is a journal article, not a grant proposal) so i suggest they rewrite (and shorten) the general discussion to emphasize what they learned and the strengths of the present study, rather than the limitations, and focus on more specific future studies that would help explain the results of this one.

Specific comments:

Page 3? “Wayfinding was coined …” -> should be “the term wayfinding was coined”

Page 10: I was unable to parse this sentence: “The screen above was after a route was traced, but before ‘f’ was pressed to signal that the route was complete.”

The graphs of route time indicated that it was very skewed. Some outlier analysis or a log transform might be appropriat.

The language in the paper is sometimes more conversational than is appropriate for a journal article, for example shortening do not to “don’t” and although to “though”

6. PLOS authors have the option to publish the peer review history of their article (what does this mean?). If published, this will include your full peer review and any attached files.

Reviewer #1: No

Reviewer #2: No

---

## [Author Response · Author response to Decision Letter 0]

27 Aug 2024

Journal Requirements:

The manuscript and manuscript files have been formatted according to the requirements above.

This research was completed under a cooperative agreement, as described in the Cover Letter and the submission portal. Although there was not a grant to a specific researcher, Heather Urry (HLU) can be identified as the recipient for ease of reporting.

3. Please note that funding information should not appear in the Acknowledgments section or other areas of your manuscript. We will only publish funding information present in the Funding Statement section of the online submission form. Please remove any funding-related text from the manuscript. 

4. Please ensure that you refer to Figure 7 in your text as, if accepted, production will need this reference to link the reader to the figure.

We revised the figures and referred to each figure in the text.

5. Please upload a new copy of Figures 1-3 as the detail is not clear. Please follow the link for more information: 

https://blogs.plos.org/plos/2019/06/looking-good-tips-for-creating-your-plos-figures-graphics/

https://blogs.plos.org/plos/2019/06/looking-good-tips-for-creating-your-plos-figures-graphics/

We revised the figures and reformatted them according to the specifications above.

6. We note that Figures 1-3 in your submission contain map images which may be copyrighted. All PLOS content is published under the Creative Commons Attribution License (CC BY 4.0), which means that the manuscript, images, and Supporting Information files will be freely available online, and any third party is permitted to access, download, copy, distribute, and use these materials in any way, even commercially, with proper attribution. For these reasons, we cannot publish previously copyrighted maps or satellite images created using proprietary data, such as Google software (Google Maps, Street View, and Earth). For more information, see our copyright guidelines: http://journals.plos.org/plosone/s/licenses-and-copyright.

1) You may seek permission from the original copyright holder of Figures 1-3 to publish the content specifically under the CC BY 4.0 license. 

We cannot obtain an exception to the Google Maps policy for these figures, so we have proceeded with option 2) below.

2) If you are unable to obtain permission from the original copyright holder to publish these figures under the CC BY 4.0 license or if the copyright holder’s requirements are incompatible with the CC BY 4.0 license, please either i) remove the figure or ii) supply a replacement figure that complies with the CC BY 4.0 license. Please check copyright information on all replacement figures and update the figure caption with source information. If applicable, please specify in the figure caption text when a figure is similar but not identical to the original image and is therefore for illustrative purposes only.

We chose to remove the copyrighted content from the publication and did not create replacement content. Instead, we have retained the most relevant details and referred to content that is available for fair use on OSF.

Captions for the three supplemental materials have been added at the end of the manuscript.

Reviewer #1: 

This is a well written paper which addresses and interesting question and render some interesting results. Having said this, I have a couple of major concerns which I will outline below. I believe that these concerns need to be addressed before the paper can be published.

Thank you. We believe that we have addressed each of the concerns in the revised manuscript.

Major concerns:

The authors state that the urgency manipulation affect planning time and planning performance. At this point, I have a couple of doubts: My main concern is that participants do not need to have a complete plan at the end of the planning phase, but that they feel urged by the instructions to end the planning phase without having a fully formed plan. Instead they continue planning or refining their route plan while they are tracing the route. So, an alternative explanation of their findings is that participants only form coarse plans under time pressure, rather than sub-optimal plans, which they refine during the tracing phase. 

We agree with this possible description of the planning during this task. Since we consider both aspects of the task route planning and we don’t have the ability to distinguish between coarse or suboptimal plans, we have re-framed the message to better convey the interesting effects of urgency. The effect of urgency on route length (the portion which could involve refining of a plan) was conditional on subjective stress, but it was the same direction as the effect of urgency on planning time (i.e., less planning time in addition to shorter routes traced).

Therefore, we can conclude that in the alternative explanation, even if urgency resulted in more coarse plans in the planning time phase, it also resulted in shorter routes traced during the tracing phase, which could be a result of cognitive processes or fine motor performance. We did not report on the time spent tracing, but the tracing time was positively correlated with route length, so if shorter routes were due to more plan refinement, refinement during the tracing phase might be negatively correlated with tracing time. We did not speculate on this alternative explanation in the manuscript.

This might also explain why routes under time pressure instructions were less efficient than without explicit instructions. 

We didn’t test route efficiency, only route length, which does not account for the optimal route length. We did test ideal route length (“as the crow flies”) as a covariate in exploratory analyses, which did not change the conclusions from the confirmatory tests.

I am not convinced that the authors can address this point unless they run additional experiments, as the current procedure does not allow them to assess how refined route plans are at the end of the planning phase.

This is a limitation of the method, but it does not preclude interpretation of the results. We changed the language in the manuscript to better reflect our conceptualization of route planning, which could include plan refinement. The changing of plans during planning or during locomotion is an interesting direction for future research, but we did not highlight that idea in the revised manuscript.

Description of the task and materials: The authors should report in detail how the map stimuli were designed, what criteria they used to design the maps, how many alternative routes between start and destination were (reasonably) available, etc.

We describe the process in more detail in the revised manuscript, and also provide a description here.

The maps were created by taking screenshots of google maps. The location for each map stimulus was relatively random. A graduate student looked for areas that were in or adjacent to cities or towns, such that there were multiple ways to get from place to place. Each map had at least two routes that were roughly equivalent, although one could be slightly shorter. There were also potential cognitive differences (e.g., one route option might have fewer turns). There were often more than two possible routes, but we noted at least two routes while creating the stimuli.

The authors also mention that some map stimuli had additional waypoints. Depending on where the waypoints were positioned, they could either reduce or increase the difficulty of the planning task and affect the number of reasonable alternatives participants need to consider. Overall, the description of the materials and methods should be improved. For example, the authors need to explain how participants were instructed - were they instructed to plan the shortest possible route?

We made sure that the participant instructions were clearly described in the manuscript. Participants were instructed to complete as many routes as possible and motivated by a small performance bonus for each route completed successfully. These instructions could imply that they should plan shorter or more efficient routes, but the explicit instruction was only to complete as many routes as possible, without tracing errors, in the allotted time. 

Detailed comments:

The authors refer to their task as a wayfinding task. I believe that the community would expect wayfinding tasks to have an element of actual navigation/movement through space, rather than just planning routes from maps. I would therefore suggest that the authors think about the terminology they use as the address one aspect of wayfinding with maps, i.e. the initial planning stage from maps, but their work cannot speak to how stress might affect the execution of the plan (which is more than just locomotion as it also required memory and decision making to actually follow the route). This is further supported by Lynch’s quote “the process of determining and following a path or route between an origin and a destination”.

We agree that our study is not wayfinding that includes large-scale navigation/movement through space. However, there is an element of behavioral performance in the tracing portion of the task, so we hesitate to draw a hard line between wayfinding in this task and wayfinding in the real world, beyond the admittedly very clear distinction between fine and gross motor performance. To highlight the importance of our task, and also avoid confusion with other wayfinding tasks, we have replaced the word “wayfinding” with the more precision phrase “route planning” when there could be ambiguous interpretations. We also avoided referring to “aided wayfinding”, since that could also imply something other than route planning.

I find it difficult to reconcile H0 and H2 which seem to contradict each other: H0 states that less time would result in fewer trials, while H2 states that less time should result in quicker planning, i.e. shorter trials, which should increase the number of trials. This should be clarified. I assume this has to do with substantially longer waiting times than planning times, however, this would then render H0 trivial.

Time pressure is complicated. There can be a total time limit, which could simultaneously decrease the “amount of task completed” (e.g., the distance travelled or the number of map trials completed), but also increase the speed at which the task is completed (e.g., driving/walking/biking speed, or planning and tracing time). We clarified in the document that planning time in H2 refers to the time between stimulus presentation and the beginning of tracing, which occurs on every trial completed by each participant.

I am not sure if this relates to a requirement of the journal in which case I would suggest to redirect this comment to the editor, but I much prefer having figures embedded in the text. This makes it so much easier for reviewers to read and assess the paper as there is no need to endlessly scroll back and forth through the document.

This is a requirement of the journal.

Do you have any information about the interaction device participants used to complete the task. I would imagine that completing the task with a mouse or trackpad could substantially affect the times required to trace a route.

We programmed the experiment to detect touchscreen use (to increase confidence in exclusions for touchscreen use) and participants also self-reported device use. Based on this feedback, we tested self-reported trackpad compared to mouse use as a covariate in exploratory models, reported in S1_File, Table S5. The difference was not statistically significant (H1-H3) unless the models controlled for subjective stress (H4-H5). Including the covariate did not alter conclusions and we mentioned this in the manuscript. 

I am unclear what the following text means: “A map stimulus was presented at the start of each trial. The cursor appeared as a blue dot when ‘s’ was pressed to signal the end of the planning phase. The screen above was after a route was traced, but before ‘f’ was pressed to signal that the route was complete.”

We revised the description to avoid unnecessary detail like the appearance of the cursor as a blue dot while conveying the relevant information about the trial procedure.

Fig 3 suggests that it may take some time to find the start and destination location which were only two symbols among a lot of additional markers in the maps and it is unclear what a waypoint is (referred to in the text but not explained in detail). The screenshots of the maps stimuli are of low image quality which makes it difficult to read the maps and therefore understand the task. Was the time to find the start location, the destination and potential waypoints included in the planning phase?

We revised the manuscript to convey that location identification is conceptually part of the planning time, but a participant could have ended the planning phase and began the tracing phase without locating all the points. We also removed screenshots of the maps seen by participants from the publication due to licensing restrictions, while providing them for fair use in the online repository.

The authors state that the planning phase is distinct from the tracing phase. What prevents participants from moving into the tracing phase without having completed the plan and using time in t

---

## [Decision Letter · Decision Letter 1]

29 Sep 2024

PONE-D-23-34568R1How much time to figure out how to get where? Route planning and subjective stress under time pressurePLOS ONE

Dear Mr. Plonski,

Thank you for submitting your revised manuscript to PLOS ONE.

I have now received the comments on your revised manuscript from the two original reviewers – please find their reviews at the end of this letter.

As you will see, both reviewers appreciated your efforts in addressing most of their original points of critique. However, as Reviewer 1 outlined, there are several outstanding issues around the methodology and data analysis of the study, as well as the study conclusions, that require further attention. Therefore, I would like to invite you to submit a revised version of the manuscript that addresses all the points raised by the latest round of reviews.

We look forward to receiving your revised manuscript.

Kind regards,

Ioanna Markostamou, Ph.D.

Academic Editor

PLOS ONE

**Reviewers' comments:**

Reviewer's Responses to Questions

**Comments to the Author**

1. If the authors have adequately addressed your comments raised in a previous round of review and you feel that this manuscript is now acceptable for publication, you may indicate that here to bypass the “Comments to the Author” section, enter your conflict of interest statement in the “Confidential to Editor” section, and submit your "Accept" recommendation.

Reviewer #1: (No Response)

Reviewer #2: All comments have been addressed

2. Is the manuscript technically sound, and do the data support the conclusions?

Reviewer #1: Partly

Reviewer #2: Yes

3. Has the statistical analysis been performed appropriately and rigorously? 

Reviewer #1: Yes

Reviewer #2: Yes

4. Have the authors made all data underlying the findings in their manuscript fully available?

Reviewer #1: Yes

Reviewer #2: Yes

5. Is the manuscript presented in an intelligible fashion and written in standard English?

Reviewer #1: Yes

Reviewer #2: (No Response)

6. Review Comments to the Author

**Reviewer #1**: The authors have addressed most of my original comments.

Figures: I cannot find the figures in the revised manuscript which makes it difficult to assess several of the revisions.

Map Stimuli: While the process of generating the maps has improved in the revision, I believe the rationale for having three categories or sets of maps should be further detailed. In addition, I would suggest to present at least one examples for each map type (see comment above about Figures not showing in the submitted revision)

Methods: I cannot find information about how the maps were presented to participants. Were maps randomly assigned to the different conditions or was this balanced between participants?

Route Planning Task & Motor Performance: I am still a little bit puzzled about the planning task, especially related to the tracing aspect and the required motor performance. Why was the decision made to ask participants to trace the chosen/planned path. How does that aspect of the task relate to the real world examples, the authors introduce?

Route Planning Task & Motor Performance: I understand that motor performance is not a focus on the study, but I assume that time pressure also affected motor performance in the tracing task. Is this true? How many tracing errors were there and how many trials were discarded because participants did not meet the motor performance expectations?

Route Planning Task & Motor Performance: How long, on average and for the different conditions, did it take participants to trace the routes? Overall, I find it surprising that the tracing aspect and the motor skills required to trace the routes is such an important part of the task but is not addressed in the analysis at all.

In their response to reviewer comments the authors argue that the search for the start/destination/waypoints is part of the planning process. It would be good to see more example stimuli to develop a better understanding of how long it might take to find these symbols to get a better idea of the relative impact of the different processes (search, actual path planning) on overall planning time.

In their response to reviewer comments about outliers in the plan time analysis, the authors respond that they “excluded trials for confirmatory hypothesis testing based on the preregistered criteria.”. I am not sure what that means. Have they actually excluded outliers? I believe this is an important point as the data looks very skewed.

Route length: I do understand that the authors analyse route lengths but I am not sure I understand what exactly participants were instructed to do. Were they instructed to plan short routes (I assume this is the case otherwise it is unclear why the authors chose the route length as a DV). If the authors explicitly instructed participants to plan short routes, why was the chosen route not compared to the optimal route which seems the most logic measure (unless of course, participants planned routes that were easy to trace - see point above)? In their response to a similar earlier comment, the authors just stated that they chose two-dimensional distance and that other measures could be looked at but they do not motivate why they choose distance and what it tells us about the route planning process.

Route length & Route tracing: Related to the points raised above, I would be interested to know if the authors agree that the specific procedure and the requirement to precisely trace the chosen routes could influence route choice? Without having performed the task myself, I suspect that tracing a route closely is difficult. If so, this requirement could well affect route choice, such as choosing the simplest route. If that was the case, the results are strongly affected by the response mode, and one could question what the results tell us about route planning itself.

Analysis: Why was map set (no waypoints, waypoints, round trip) not included in the analysis as a factor. This relates to an earlier comment about the missing motivation for including the different map types.

**Reviewer #2**: (No Response)

7. PLOS authors have the option to publish the peer review history of their article (what does this mean?). If published, this will include your full peer review and any attached files.

Reviewer #1: No

Reviewer #2: No

---

## [Author Response · Author response to Decision Letter 1]

13 Nov 2024

Reviewer #1: The authors have addressed most of my original comments.

Figures: I cannot find the figures in the revised manuscript which makes it difficult to assess several of the revisions.

We are not sure why the documentation provided to the reviewer seemed not to contain the figures. The figures are not embedded in the manuscript, per the journal requirements, but they were visible to us, starting on p. 54 of the manuscript pdf generated by the submission portal. 

Map Stimuli: While the process of generating the maps has improved in the revision, I believe the rationale for having three categories or sets of maps should be further detailed. In addition, I would suggest to present at least one examples for each map type (see comment above about Figures not showing in the submitted revision)

We created multiple categories of map, based on the placement of the points, because we wanted to be sure there was variation in the kinds of routes participants had to plan and trace. We improved the description of this rationale in Materials/Map Stimuli.

Map type was not a variable we intended to focus on as a manipulated factor in analyses. We did, however, explore the type of map in the Supplemental Material. Adding the map type as a covariate did not change our confirmatory results, but there were statistically significant, yet unsurprising differences in the performance metrics based on map type.

All map stimuli are available at <https://osf.io/w36xe/>. We uploaded the .PNG files separately so that one does not need to download the entire task folder to access any single map image. The map stimuli themselves are not included in the document due to copyright restrictions. The file that shows the coordinates (S1_Map_Coordinates.csv) has been updated to include a column showing which of the three types of map each stimulus represents. 

Methods: I cannot find information about how the maps were presented to participants. Were maps randomly assigned to the different conditions or was this balanced between participants?

For each participant, one map was randomly assigned at the beginning of each trial, without replacement and without regard for any between- or within-persons condition. This has been clarified in the Procedure (L407-409).

Route Planning Task & Motor Performance: I am still a little bit puzzled about the planning task, especially related to the tracing aspect and the required motor performance. Why was the decision made to ask participants to trace the chosen/planned path. How does that aspect of the task relate to the real world examples, the authors introduce?

The tracing part of the task enabled us to assess route length as one aspect of motor performance. In general, we acknowledge that this laboratory-style task is, of course, different in lots of ways from real-world examples. This new procedure nonetheless leaves future research to explore how the route length outcome from this task is associated with characteristics, including lengths, of routes planned or navigated in real-world situations.

Route Planning Task & Motor Performance: I understand that motor performance is not a focus on the study, but I assume that time pressure also affected motor performance in the tracing task. Is this true? How many tracing errors were there and how many trials were discarded because participants did not meet the motor performance expectations?

We think of route length as one indicator of fine motor performance because, while the route length is highly determined by the stimulus (and this is accounted for in the models), there may be some variability in route length related to the ability of participants to complete the task with their hand. The two-dimensional route length was the only outcome that involved fine motor performance we conceptualized in our preregistration for this study, including the hypotheses and analysis plan. However, like the reviewer, we are very interested in other outcomes. In fact, we are currently pursuing publication of a separate manuscript that examines a wider range of motor performance outcomes based on data collected in additional studies. 

We clarified the exclusion procedure in this revision, to better convey that exclusions took place at the participant level first. Based on the pre-registered performance criteria, 19% of the participants were excluded because, on at least five trials, the data indicated that they either 1) did not click the right-hand button to start the trial, 2) had a planning time greater than 60 seconds, or 3) did not attempt to follow the paths and draw a route through all the marked points. Only 18 trials (<1%) were excluded due to the performance review flag, after the participant-level exclusions.

Route Planning Task & Motor Performance: How long, on average and for the different conditions, did it take participants to trace the routes? Overall, I find it surprising that the tracing aspect and the motor skills required to trace the routes is such an important part of the task but is not addressed in the analysis at all.

In this manuscript, we chose to focus on the pre-registered, confirmatory hypotheses. However, we have added an additional exploratory model to the main text that uses the outcome of “tracing time”. We expected that tracing time (M = 35.93 s) would be correlated with route length, so much so that we thought it may be redundant (i.e., if a route is longer, it should take longer to trace that route). Although the correlation was large, r = .56, it was not as large as we expected a priori. Tracing time results were added to Tables 2-4 in the main text. 

The distribution of the tracing time metric was similar to the planning time metric, so we modelled it as a non-normally-distributed outcome, using a generalized linear mixed-effect model with a gamma distribution and a log link. The exploratory model with the tracing time outcome provides additional information that should supplement the pre-registered performance outcomes (i.e., planning time, route length) that we conceptualized for confirmatory hypothesis testing.

In their response to reviewer comments the authors argue that the search for the start/destination/waypoints is part of the planning process. It would be good to see more example stimuli to develop a better understanding of how long it might take to find these symbols to get a better idea of the relative impact of the different processes (search, actual path planning) on overall planning time.

As mentioned above, all map stimuli are available at <https://osf.io/w36xe/>.

In their response to reviewer comments about outliers in the plan time analysis, the authors respond that they “excluded trials for confirmatory hypothesis testing based on the preregistered criteria.”. I am not sure what that means. Have they actually excluded outliers? I believe this is an important point as the data looks very skewed.

We did not plan to exclude outliers based on a statistical cutoff like a z-score. Instead, we carefully considered the conditions before data collection and made a priori decisions about the conditions under which the data would be invalid (e.g., a planning time greater than 60 seconds). When the planning time outcome was not normally distributed, we changed our analysis plan and applied a gamma distribution with a log link to address that specific violation of distributional assumptions when modeling planning time (see Results, Assumption Check). 

We excluded one participant and one trial for unexpected reasons that were clearly justifiable (one participant essentially did not complete the task, so had one extraordinarily long planning time, and, for another participant, one trial had a planning time that was erroneously zero). Although it may be tempting to test outlier exclusion procedures, it is important to have a valid reason for doing so, because removing outliers only to improve model fit could lead to invalid conclusions. However, to address the reviewer’s point directly, we conducted a test of influential observations for all models reported in the main text. We examined Cook’s distance as an arbitrary metric that indexes the change in model estimates when observations are excluded from the model, one by one. 

When we re-ran the models excluding observations with a Cook’s distance < .5, the results did not alter our main conclusions about urgency messaging and subjective stress. The statistical significance of three fixed-effect predictors was affected. The changes in statistical significance were only observed in the generalized linear mixed-effects models that used non-normal distributions, for which the Cook’s distance may not make sense. After excluding 2 influential observations in the model for H4, the estimated effect of long waits compared to short waits was no longer statistically significant (OR = 1.115, p = .030 to OR = 1.133, p = .159). After excluding 3 influential observations in the model for H4E, the estimated effect of long waits compared to short waits was no longer statistically significant (OR = 1.101, p = .001 to OR = 1.110, p = .173) and neither was the association between subjective stress at the individual-level and tracing time (b = 0.001, p < .001 to b = 0.003, p = .096).

Route length: I do understand that the authors analyse route lengths but I am not sure I understand what exactly participants were instructed to do. Were they instructed to plan short routes (I assume this is the case otherwise it is unclear why the authors chose the route length as a DV). If the authors explicitly instructed participants to plan short routes, why was the chosen route not compared to the optimal route which seems the most logic measure (unless of course, participants planned routes that were easy to trace - see point above)? In their response to a similar earlier comment, the authors just stated that they chose two-dimensional distance and that other measures could be looked at but they do not motivate why they choose distance and what it tells us about the route planning process.

Thank you for giving us the opportunity to clarify. We shared our reasoning regarding a focus on route length on lines 172-176, where we said, "Finally, because time pressure generally increases speed and decreases accuracy we expected that the higher-time pressure conditions would result in longer routes than the lower-time pressure conditions (H3). However, because the effects of acute time pressure in new map environments have not been studied extensively, we intentionally remained open to opposite or null effects of all three experimental manipulations.” We note on lines 286-287 that "Instructions emphasized tracing as many routes as possible without making tracing errors." 

In general, we expected that the two outcomes we selected would provide useful information about performance in this task, which was designed as a controlled laboratory-style experiment that would allow participants to plan routes in previously unlearned environments. We expected the performance metrics to be affected by the manipulated temporal conditions. However, we do not claim that this task directly assesses how participants plan routes in real life.

Route length & Route tracing: Related to the points raised above, I would be interested to know if the authors agree that the specific procedure and the requirement to precisely trace the chosen routes could influence route choice? Without having performed the task myself, I suspect that tracing a route closely is difficult. If so, this requirement could well affect route choice, such as choosing the simplest route. If that was the case, the results are strongly affected by the response mode, and one could question what the results tell us about route planning itself.

Agreed, the task enables participants to make route choices. It's entirely plausible that route choices are impacting planning time and route length outcomes. The rational move under time pressure is to indeed try to draw simple routes so they take less time to plan and trace. This will definitely be useful to explore in future studies. Interestingly, only about 9% of variation in route lengths is due to participants whereas a whopping 69% of variation in route lengths is due to the different map stimuli. So the routes people trace are largely due to features of the maps themselves; some maps require longer routes than others.

Analysis: Why was map set (no waypoints, waypoints, round trip) not included in the analysis as a factor. This relates to an earlier comment about the missing motivation for including the different map types.

We included the map type as a predictor in the models reported in the Supplemental Material. The interpretation of the models used for confirmatory hypothesis testing did not change when we accounted for the type of map; however, map type was associated with some outcomes. Furthermore, in the current revision, we included both map type and variation in the length of the ideal routes (i.e., as the crow flies) and the interpretation of the models used for confirmatory hypothesis testing did not change. However, ideal route length was also associated with some performance outcomes.

---

## [Decision Letter · Decision Letter 2]

11 Dec 2024

How much time to figure out how to get where? Route planning and subjective stress under time pressure

PONE-D-23-34568R2

Dear Dr. Plonski,

Thank you for submitting your revised manuscript to PLOS ONE.

The reviewers and I have read through your revised paper and response to reviewers’ comments and we all agree that you have appropriately addressed all issues raised in your revised manuscript. No further points of critique were raised.

I believe that the two rounds of revisions have improved the clarity of the paper and enhanced its potential impact. Therefore, I am happy to confirm that your manuscript has been judged scientifically suitable for publication and will be formally accepted for publication once it meets all outstanding technical requirements.

Kind regards,

Ioanna Markostamou, Ph.D.

Academic Editor

PLOS ONE

Additional Editor Comments (optional):

Reviewers' comments:

Reviewer's Responses to Questions

**Comments to the Author**

1. If the authors have adequately addressed your comments raised in a previous round of review and you feel that this manuscript is now acceptable for publication, you may indicate that here to bypass the “Comments to the Author” section, enter your conflict of interest statement in the “Confidential to Editor” section, and submit your "Accept" recommendation.

Reviewer #1: All comments have been addressed

2. Is the manuscript technically sound, and do the data support the conclusions?

Reviewer #1: Yes

3. Has the statistical analysis been performed appropriately and rigorously? 

Reviewer #1: Yes

4. Have the authors made all data underlying the findings in their manuscript fully available?

Reviewer #1: Yes

5. Is the manuscript presented in an intelligible fashion and written in standard English?

Reviewer #1: Yes

6. Review Comments to the Author

Reviewer #1: I do not have any further suggestions or comments.

From my comments it should be clear that I have a couple of smaller reservations, particularly about the response mode and what it can tell us about route choice/route planning. However, the authors addressed my comments/suggestions carefully and explained that they are in the process of addressing some of the issues I raised in further studies/publications. I therefore recommend publication of the study and I am looking forward to learning more about future studies on this topic.

7. PLOS authors have the option to publish the peer review history of their article (what does this mean?). If published, this will include your full peer review and any attached files.

Reviewer #1: No

---

## [Editor Report · Acceptance letter]

22 Dec 2024

PONE-D-23-34568R2 

PLOS ONE

Dear Dr. Plonski, 

I'm pleased to inform you that your manuscript has been deemed suitable for publication in PLOS ONE. Congratulations! Your manuscript is now being handed over to our production team.

Kind regards, 

on behalf of

Dr. Ioanna Markostamou 

Academic Editor

PLOS ONE